JCB Journal of Cell Biology

# The V-ATPase–ATG16L1 axis recruits LRRK2 to facilitate the lysosomal stress response

Tomoya Eguchi[1,2]*, Maria Sakurai[1]*, Yingxue Wang[3], Chieko Saito[2], Gen Yoshii[1], Thomas Wileman[3], Noboru Mizushima[2], Tomoki Kuwahara[1], and Takeshi Iwatsubo[1]

**Leucine-rich repeat kinase 2 (LRRK2), a Rab kinase associated with Parkinson's disease and several inflammatory diseases, has been shown to localize to stressed lysosomes and get activated to regulate lysosomal homeostasis. However, the mechanisms of LRRK2 recruitment and activation have not been well understood. Here, we found that the ATG8 conjugation system regulates the recruitment of LRRK2 as well as LC3 onto single membranes of stressed lysosomes/phagosomes. This recruitment did not require FIP200-containing autophagy initiation complex, nor did it occur on double-membrane autophagosomes, suggesting independence from canonical autophagy. Consistently, LRRK2 recruitment was regulated by the V-ATPase–ATG16L1 axis, which requires the WD40 domain of ATG16L1 and specifically mediates ATG8 lipidation on single membranes. This mechanism was also responsible for the lysosomal stress-induced activation of LRRK2 and the resultant regulation of lysosomal secretion and enlargement. These results indicate that the V-ATPase–ATG16L1 axis serves a novel non-autophagic role in the maintenance of lysosomal homeostasis by recruiting LRRK2.**

## Introduction

Mutations in the leucine-rich repeat kinase 2 (*LRRK2*) gene are the most common risk alleles for autosomal-dominantly inherited Parkinson's disease (PD) as well as one of the top risk alleles for sporadic PD (Trinh and Farrer, 2013). In addition, *LRRK2* is associated with several inflammatory or infectious diseases, including Crohn's disease, systemic lupus erythematosus, and leprosy (Barrett et al., 2008; Trinh and Farrer, 2013; Zhang et al., 2009, 2017). *LRRK2* encodes a large protein kinase that phosphorylates a subset of Rab GTPases (e.g., Rab8, Rab10) (Steger et al., 2016, 2017) and is expressed in various cell/tissue types, including neurons, glial cells, peripheral cells, and immune cells.

Recent studies have suggested that LRRK2 regulates lysosomal functions in both physiological and pathological contexts. Inhibition or loss of LRRK2 in vivo has been shown to cause the accumulation of enlarged secondary lysosomes or lysosome-related organelles, especially in the kidney and lung (Herzig et al., 2011; Tong et al., 2010). Abnormal lysosome morphology has also been observed in cells harboring *LRRK2* familial mutations (Henry et al., 2015; Hockey et al., 2015; Kuwahara and Iwatsubo, 2020). Such lysosomal abnormalities have attracted much attention as possible causes of PD (Abe and Kuwahara, 2021).

We have shown previously that the incubation of cells with lysosomotropic agents or proton ionophores, for example,

chloroquine (CQ) and monensin, causes LRRK2 recruitment onto the stressed lysosomes. LRRK2 on lysosomes phosphorylates Rab8/10 and leads to the promotion of lysosomal enzyme secretion and the suppression of lysosomal enlargement (Eguchi et al., 2018; Kuwahara et al., 2020). Other groups have reported that the lysosomal membrane-damaging agent L-Leucyl-L-Leucine methyl ester (LLOMe) triggers lysosomal recruitment and activation of LRRK2 (Bonet-Ponce et al., 2020; Herbst et al., 2020), although its ability to activate LRRK2 might be weaker than that caused by CQ or proton ionophores (Kalogeropulou et al., 2020). Therefore, several types of lysosomal stresses are thought to cause the lysosomal recruitment and local activation of LRRK2, although it is still unclear how lysosomal stresses are sensed and activate LRRK2.

Recent studies have shown that lysosomotropic agents and proton ionophores, including CQ and monensin, induce the recruitment of LC3 (one of the ATG8 family proteins), a well-known autophagosome marker, onto single-membrane lysosomes (Florey et al., 2015). This recruitment onto single membranes is specifically mediated by the ATG8 conjugation system composed of several ATG proteins, including ATG5, ATG7, and ATG16L1, and is recently termed "CASM" (conjugation of ATG8 to endolysosomal single membranes) (Durgan and Florey, 2022). Similar ATG8 recruitment has also been

[1]Department of Neuropathology, Graduate School of Medicine, The University of Tokyo, Tokyo, Japan;   [2]Department of Biochemistry and Molecular Biology, Graduate School of Medicine, The University of Tokyo, Tokyo, Japan;   [3]Norwich Medical School, University of East Anglia, Norwich, UK.

*T. Eguchi and M. Sakurai contributed equally to this paper.   Correspondence to Tomoki Kuwahara: kuwahara@m.u-tokyo.ac.jp;   Takeshi Iwatsubo: iwatsubo@m.u-tokyo.ac.jp.

observed on single-membrane phagosomes of phagocytic cells that have engulfed extracellular particles (Sanjuan et al., 2007), and this is specifically termed "LC3-associated phagocytosis (LAP)." Mechanistically, the binding of ATG8 to single membranes does not involve the autophagy initiation complex composed of ULK1/2, FIP200, ATG13, and ATG101 and appears to be independent of canonical autophagy. Interestingly, ATG8 recruitment to single membranes is dependent on the WD40 domain of ATG16L1, which is dispensable for the canonical macroautophagy (Fletcher et al., 2018; Rai et al., 2019; Wang et al., 2021). Further studies revealed that the WD40 domain of ATG16L1 directly interacts with the vacuolar H(+)-ATPase (V-ATPase) on lysosomal membranes to recruit the ATG8 conjugation system (Hooper et al., 2022; Ulferts et al., 2021; Xu et al., 2019), and this "V-ATPase–ATG16L1 axis" is now thought to be at the core of CASM. However, the precise roles of ATG8 conjugation to single membranes via this axis are still largely unknown.

Here, we show that upon lysosomal stresses, LRRK2 is co-recruited with LC3 onto single-membrane lysosomes via the ATG8 conjugation system. This recruitment was not mediated by the autophagy initiation complex but required the WD40 domain of ATG16L1, suggesting the involvement of the V-ATPase–ATG16L1 axis. This axis also regulated the activation of LRRK2, allowing the release of lysosomal enzymes and suppression of lysosomal enlargement. These results suggest a novel role of the V-ATPase–ATG16L1 axis in lysosomal homeostasis via LRRK2 recruitment.

## Results

### LRRK2 is recruited onto single-membrane lysosomes together with LC3

We have previously observed that phagosomes or stressed lysosomes in RAW264.7 cells recruit not only LRRK2 but also LC3 (Eguchi et al., 2018). To more clearly define the relationship between LRRK2 and LC3 recruitment to these compartments, RAW264.7 cells were treated with CQ to swell lysosomes or zymosan (a yeast cell wall preparation) to identify phagosomes. The recruitment of endogenous LRRK2 was analyzed separately for LC3-positive and LC3-negative lysosomes/phagosomes. The results showed that LRRK2 was selectively localized on the LC3-positive lysosomes and phagosomes (Fig. 1, A and B). The specificity of immunostaining with anti-LRRK2 antibody was confirmed using *Lrrk2* knockout (KO) RAW264.7 cells, where the signal of the antibody was not observed (Fig. S1 A). As LC3 is a well-known marker of double-membrane autophagosomes formed during autophagy, we also examined whether LRRK2 is recruited onto LC3-positive autophagosomes that are induced by starvation. LC3 puncta indicative of autophagosomes were detected in the cytoplasm, but these rarely colocalized with LRRK2 (Fig. 1 C).

It has been shown that LRRK2 is recruited onto lysosomes in cells treated with LLOMe, which induces lysosomal rupture and causes lysophagy, a type of macroautophagy that degrades damaged lysosomes by engulfing them in LC3-positive double-membrane autophagosomes (Bonet-Ponce et al., 2020; Herbst et al., 2020). To clarify whether LRRK2 is colocalized with LC3-positive structures during lysophagy, cells incubated with

LLOMe or CQ were stained for galectin-3, a marker for ruptured lysosomes. The maximum duration of LLOMe treatment was 1 h because of its high cytotoxicity. In contrast to LLOMe treatment that strongly increased galectin-3–positive ruptured lysosomes, CQ treatment did not efficiently increase galectin-3–positive lysosomes, even though CQ treatment induced lysosomal targeting of LRRK2 more efficiently than LLOMe (Fig. 1, D and E). These data suggest that LRRK2 recruitment to LC3-positive lysosomes does not represent the response to lysophagy. This is consistent with the previous reports that lysophagy does not occur on LRRK2-positive lysosomes damaged by LLOMe (Bonet-Ponce et al., 2020; Herbst et al., 2020).

Recent studies have shown that LC3 is conjugated to the single membrane of phagosomes, following uptake of extracellular particles, to lysosomes stressed by CQ or to ruptured lysosomes causing proton leakage (Cross et al., 2023; Florey et al., 2015; Heckmann et al., 2017). To determine whether LRRK2–LC3 double-positive structures are composed of single membranes or double membranes, we analyzed these structures by correlative light and electron microscopy (CLEM), which enables direct superposition of optical and electron microscopic images. CLEM analysis of HEK293 cells coexpressing GFP-LC3 and mCherry-LRRK2 revealed that LRRK2 and LC3 were colocalized on the single-membrane structures (Fig. 1 F and Fig. S1, B and C), whereas typical autophagosomes and nuclear membranes were observed as double-membrane structures. These data further support the notion that even though the stressed lysosomes recruit LC3, the targeting of LRRK2 to lysosomes does not involve canonical autophagy.

### Recruitment of LRRK2 is mediated by the ATG8 conjugation system but not by the autophagy initiation complex

Colocalization of LRRK2 and LC3 suggests that they may work together to control lysosome homeostasis. Because previous studies have implied that autophagy can be regulated by LRRK2 (Madureira et al., 2020), we tested whether LRRK2 also regulates the recruitment of LC3 to stressed lysosomes. However, LRRK2 knockdown did not suppress the increase in LC3-II (lipidated LC3) induced upon CQ treatment (Fig. 2, A and B), indicating that LRRK2 is not upstream of LC3 lipidation. Then, we tested the possibility that components of the ATG8 conjugation system that operates during autophagy may also regulate the lysosomal targeting of LRRK2. ATG8 is a ubiquitin-like protein and is conjugated to phosphatidylethanolamine or phosphatidylserine by the combined action of the E1-like enzyme ATG7, the E2-like enzyme ATG3, and the E3-ligase-like activity of the ATG5-ATG12:ATG16L1 complex. Knockdown of ATG5, ATG7, or ATG16L1 in RAW264.7 cells significantly suppressed the lysosomal targeting of LRRK2 in response to CQ treatment (Fig. 2, C and E). Manders' colocalization coefficient (MCC) of LRRK2 and LAMP1 was also increased upon CQ treatment, although the increase was small possibly due to the remaining cytoplasmic pool of LRRK2, and the increase in MCC was canceled by knockdown of ATG5 (Fig. S2 A). The ATG5 dependency in the lysosomal targeting of LRRK2 was further confirmed by biochemical analysis, where lysosomes internalizing iron-dextran were magnetically isolated and LRRK2 in this

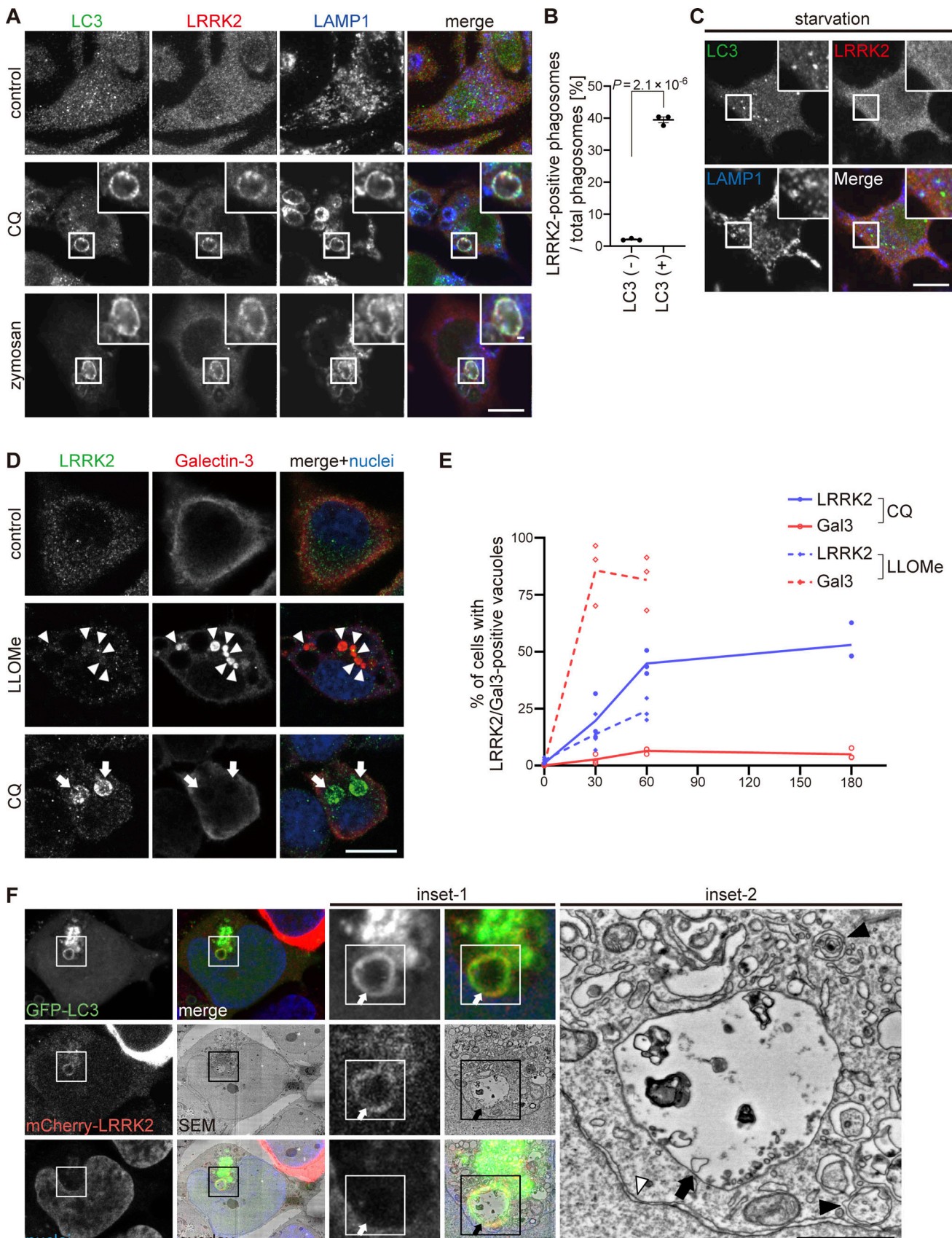

**Figure 1. LRRK2 colocalized with LC3 on lysosomal single membranes. (A)** Fluorescence images of endogenous LRRK2 and LC3 in cells in the absence of any stresses (upper panels), treated with CQ (middle panels) or with zymozan (lower panels). Scale bars, 10 and 1 μm (insets). **(B)** Percentages of LRRK2-

positive phagosomes in LC3-positive and -negative phagosomes. Data represent mean ± SEM (N = 3 independent experiments). The difference was analyzed using an unpaired two-tailed *t* test. **(C)** Fluorescence images of RAW264.7 cells cultured in amino acid–depleted medium for 90 min. **(D)** Fluorescence images of RAW264.7 cells treated with LLOMe or CQ. Arrowheads indicate Gal3-positive and LRRK2-negative lysosomes and arrows indicate LRRK2-positive and Gal3-negative lysosomes. Scale bar, 10 μm. **(E)** Percentage of cells harboring LRRK2-positive or Gal3-positive vacuoles over time under CQ or LLOMe treatment, as shown in D. Data represent mean ± SEM (N = 4 independent experiments, 30–74 cells were analyzed in each experiment). **(F)** CLEM analysis of LRRK2-LC3 double-positive structures. Scale bars, 10 μm, 1 μm (inset-1, inset-2). Arrows: LRRK2-LC3 double-positive membrane, black arrowhead: autophagosome, white arrowhead: nuclear envelope.

fraction was analyzed. Consistent with our immunocytochemical data, LRRK2 was highly enriched in the isolated lysosomal fraction upon CQ treatment, and this was suppressed by knockdown of ATG5 (Fig. S2 B). In addition to CQ treatment, knockdown of ATG5 or ATG16L1 also suppressed the targeting of LRRK2 onto phagosomes containing zymosan (Fig. 2, D and F). Given our observation that LRRK2 is localized on the single membranes of stressed lysosomes independently of autophagy and lysophagy, we hypothesized that lysosomal or phagosomal targeting of LRRK2 would be independent of the autophagy initiation complex composed of ULK1/2, FIP200, ATG13, and ATG101. Consistent with our prediction, knockdown of FIP200 in RAW264.7 cells did not suppress the lysosomal/phagosomal targeting of LRRK2 upon exposure to CQ or zymosan (Fig. 2, G–J). Likewise, knockdown of another component ATG13 did not affect the lysosomal targeting of LRRK2 upon CQ exposure (Fig. S3, A and B). The knockdown of FIP200 or ATG13 resulted in a large decrease in their protein level as well as in LC3 lipidation under the starved condition, confirming that autophagic activity was impaired (Fig. S3, C and D). Taken together, these results suggest that LRRK2 recruitment to stressed lysosomes requires the ATG8 conjugation system (ATG5-ATG12:ATG16L1) but is independent of canonical autophagy.

To reveal the mechanism by which ATG8 conjugation system regulates lysosomal targeting of LRRK2, we assessed the interaction between LC3A/B/C and LRRK2. However, coimmunoprecipitation of LC3 with LRRK2 was not observed, whereas p62, a well-known interactor of LC3, bound to LC3B (Fig. S4 A). Similarly, the interaction between LRRK2 and GABALAP family proteins was not detected (Fig. S4 B). We also analyzed the involvement of Rab29, a well-known interactor and upstream regulator of LRRK2 (Eguchi et al., 2018; Kuwahara et al., 2020; MacLeod et al., 2013; Purlyte et al., 2018). Similar to LRRK2, endogenous Rab29 was translocated onto lysosomes upon CQ treatment as reported (Komori et al., 2023), whereas knockdown of ATG16L1 did not suppress the lysosomal targeting of Rab29 (Fig. S4, C and D). This result suggests that the ATG8 conjugation system regulates LRRK2 independently of Rab29.

### The V-ATPase–ATG16L1 axis is involved in lysosomal targeting of LRRK2

Recent work has begun to identify pathways that can lead to the recruitment of ATG8 to single membranes independently of autophagy (Durgan and Florey, 2022; Wang et al., 2022). NADPH oxidase 2 (Nox2) complex and Rubicon are two known factors that induce ATG8 lipidation on phagosomes containing pathogens via producing reactive oxygen species (Martinez et al., 2015). Interestingly, knockdown of p22 (one of the components of the Nox2 complex) or Rubicon suppressed LRRK2

targeting to LAMP1-positive phagosomes containing zymosan (Fig. 3, A and B). Recruitment of ATG16L1 and the associated ATG8 conjugation system (ATG5-ATG12) to the lysosomal compartment can occur independently of autophagy when the WD40 domain in ATG16L1 binds the V-ATPase on lysosomal membranes. SopF, an effector protein of *Salmonella*, has ADP-ribosyl transferase activity able to transfer ADP ribose to the ATP6VOC subunit of the V-ATPase to inhibit binding to ATG16L1. In this way, SopF inhibits ATG8 lipidation on endo-lysosome membranes without any effects on autophagy (Hooper et al., 2022; Xu et al., 2019). We overexpressed SopF in HEK293 cells and treated them with CQ. SopF overexpression suppressed lysosomal targeting of LRRK2, implicating the recruitment of LRRK2 via the V-ATPase–ATG16L1 axis (Fig. 3, C and D). We further assessed the direct involvement of ATG16L1 WD40 domain in translocation of LRRK2 by using ATG16L1 WD40-deficient cells. Bone marrow–derived macrophages (BMDMs) were prepared from wild-type (WT) or ATG16L1-ΔWD40 mice (E230 mice), where a Pro231 stop codon in ATG16L1 prevents translation of the WD domain, and in doing so inactivates the V-ATPase–ATG16L1 axis (Fletcher et al., 2018; Rai et al., 2019). Consistent with the results of SopF expression, BMDMs from E230 mice showed decreased lysosomal targeting of LRRK2 upon CQ treatment (Fig. 3, E and F) or its phagosomal targeting upon engulfment of zymosan (Fig. 3, G and H). These data collectively indicate that the V-ATPase–ATG16L1 axis regulates the lysosomal/phagosomal targeting of LRRK2.

### The ATG8 conjugation system regulates LRRK2 phosphorylation of Rab10 under lysosomal stress

Since lysosomal targeting of LRRK2 is accompanied by an increase in the LRRK2 kinase activity to phosphorylate Rab10 (Eguchi et al., 2018), we next examined whether the ATG8 conjugation system also regulates CQ-induced increase in LRRK2 activity. We found that knockdown of ATG5 in RAW264.7 cells partially but significantly suppressed CQ-induced upregulation of Rab10 phosphorylation (Fig. 4, A and B). In contrast, knockdown of FIP200 did not suppress CQ-induced phosphorylation of Rab10, consistent with unaltered lysosomal targeting of LRRK2 under FIP200 knockdown (Fig. 4, A and B). The requirement of the WD40 domain of ATG16L1 in LRRK2 activation under lysosomal stress was further assessed using primary macrophages from E230 mice and *Atg16l1* KO mouse embryonic fibroblast (MEF) cells (Fig. 4, C–F). CQ treatment increased the phosphorylation of Rab10 in WT macrophages but not in E230 macrophages (Fig. 4, C and D). Similarly, in contrast to WT MEF cells, *Atg16l1* KO MEF cells did not show an increase in Rab10 phosphorylation upon CQ treatment (Fig. 4, E and F). The exogenous expression of full-length ATG16L1 rescued the increase

Figure 2. **The ATG8 conjugation system regulates lysosomal and phagosomal targeting of LRRK2 independently of the autophagy initiation complex.**
**(A)** Levels of lipidated LC3 (LC3-II) in RAW264.7 cells transfected with nontarget or LRRK2 siRNA and treated with CQ. **(B)** Quantification of LC3-II in cells, as

shown in A. Data represent mean ± SEM (*N* = 4 independent experiments). The difference was analyzed using one-way ANOVA with Tukey's test. ns, not significant. **(C and D)** Fluorescence images of endogenous LRRK2 in RAW264.7 cells transfected with the indicated siRNAs and treated with CQ (C) or zymosan (D). Arrows in C indicate LRRK2-positive lysosomes. Scale bars, 10 and 1 µm (inset). **(E and F)** Percentage of cells harboring LRRK2-positive lysosomes (E), as shown in C, or LRRK2-positive phagosomes (F), as shown in D. Data represent mean ± SEM (*N* = 3 [E] and *N* = 4 [F] independent experiments). The difference was analyzed using one-way ANOVA with Dunnet's test. **(G and H)** Fluorescence images of endogenous LRRK2 in RAW264.7 cells transfected with nontarget or FIP200 siRNA and treated with CQ (G) or zymosan (H). Scale bars, 10 µm. **(I and J)** Percentages of cells harboring LRRK2-positive lysosomes (I) or phagosomes (J) as shown in G and H, respectively. Data represent mean ± SEM (*N* = 10 independent experiments). The difference was analyzed using unpaired two-tailed *t* test. ns, not significant. Source data are available for this figure: SourceData F2.

### The ATG8 conjugation system regulates LRRK2 downstream functions on lysosomal maintenance

We further examined the involvement of the V-ATPase–ATG16L1 axis in the functions of LRRK2 on lysosomes. We have previously shown that, under lysosomal stress, LRRK2 activates the extracellular release of lysosomal contents and simultaneously prevents excessive lysosomal enlargement. This lysosomal regulation by LRRK2 has been shown to be mediated through phosphorylation of its substrate Rab GTPases, i.e., RAB8A, Rab8B, and Rab10 (Eguchi et al., 2018). We found that CQ-induced release of lysosomal cathepsin D (CatD) into the medium was suppressed by knockdown of not only LRRK2 and its substrate Rab GTPases but also ATG5 and ATG7 (Fig. 5, A and B). This suggests that the ATG8 conjugation system participates in the regulation of LRRK2-mediated lysosomal release. Similarly, CQ-induced release of mature cathepsin B (CatB), another lysosomal enzyme, was suppressed by knockdown of ATG5 (Fig. 5, C and D). In contrast, knockdown of the autophagy initiation complex component FIP200 had no effect on lysosomal release (Fig. 5, C and D), consistent with the lack of its effect on LRRK2 recruitment and activation. We also assessed CQ-induced enlargement of lysosomes, which has been shown to be suppressed by LRRK2 (Eguchi et al., 2018). Similarly to LRRK2, knockdown of ATG5 enhanced lysosomal enlargement upon CQ treatment, while knockdown of FIP200 did not (Fig. 5, E and F). We also confirmed that lysosomes were not enlarged under siATG5 treatment only without CQ (Fig. S5). These data collectively suggest that non-autophagic ATG8 lipidation machinery plays a key role in the regulation of LRRK2 functions on stressed lysosomes, in addition to the lysosomal targeting of LRRK2.

## Discussion

In this study, we identified the non-autophagic function of the ATG8 conjugation system as a novel upstream regulator of LRRK2 under lysosomal stress. The ATG8 conjugation system, but not the autophagy initiation complex containing FIP200, was required for lysosomal targeting and activation of LRRK2 as well as the resultant downstream regulation of stressed lysosomes. Importantly, this stress-related function was mediated through the WD40 domain of ATG16L1, which is dispensable for conventional macroautophagy. How the ATG8 conjugation system responds to lysosomal stress and employs LRRK2 is not

entirely elucidated, but recent studies have suggested that V-ATPase on lysosomal membranes senses lysosomal anomalies (e.g., vacuolar damage leading to elevated luminal pH) and recruits the ATG5-12-16L1 complex by directly binding to the WD40 domain of ATG16L1 (Hooper et al., 2022; Ulferts et al., 2021; Xu et al., 2019). Indeed, overexpression of SopF, which acts on V-ATPase and blocks their interaction with ATG16L1 WD40 domain (Xu et al., 2019), suppressed LRRK2 recruitment, suggesting that the V-ATPase–ATG16L1 axis plays a vital role in activating the lysosomal function of LRRK2. However, the downstream mechanism that connects the ATG8 conjugation system and LRRK2 is still unclear. Our attempts to detect direct binding of LRRK2 and ATG8 family proteins were not successful, and the observation that LRRK2 does not colocalize with LC3 during starvation induction further suggests that these two are not directly bound. We also focused on another candidate protein Rab29, a well-known upstream regulator of LRRK2, but Rab29 recruitment to the lysosome under CQ treatment was not affected by inhibition of the ATG8 conjugation system, suggesting that Rab29 is unlikely to be involved. The possibility that LRRK2 directly binds to the components of the V-ATPase–ATG16L1 axis or their interactors should be closely examined in the future.

The relationship between LRRK2 and macroautophagy has been repeatedly reported in studies of PD mechanisms (Madureira et al., 2020; Manzoni and Lewis, 2017), and a couple of studies showed the increase in the amount of lipidated LC3 (or ATG8 orthologues) in actual PD brains (Dehay et al., 2010; Tanji et al., 2011). However, it was not clear whether these were truly autophagic changes since many studies have assumed that autophagy was induced by simply showing the increased level of lipidated LC3, thus masking potential roles for non-autophagic pathways such as the V-ATPase–ATG16L1 axis. Indeed, past pathological, cell biological, and genetic studies have associated PD with lysosomal abnormalities, which may lead to the activation of CASM as well as the perturbation of autophagy. Another study has reported that LRRK2 localizes to enlarged lysosomes or vacuoles in brains of Lewy body disease patients (Higashi et al., 2009), raising the possibility that CASM-mediated lysosomal targeting of LRRK2 is activated in pathological conditions.

Recent studies have highlighted various non-autophagic functions of the ATG8 conjugation system, especially in the regulation of lysosomes (Lee et al., 2020; Nakamura et al., 2020), immune systems (Heckmann et al., 2017), and control of pathogens (Wang et al., 2021, 2022). Our findings provide further evidence for the involvement of the ATG8 conjugation system in the maintenance of lysosomes. Interestingly, both *LRRK2* and *ATG16L1* are identified as risk genes for inflammatory diseases, including systemic lupus erythematosus (Zhang et al.,

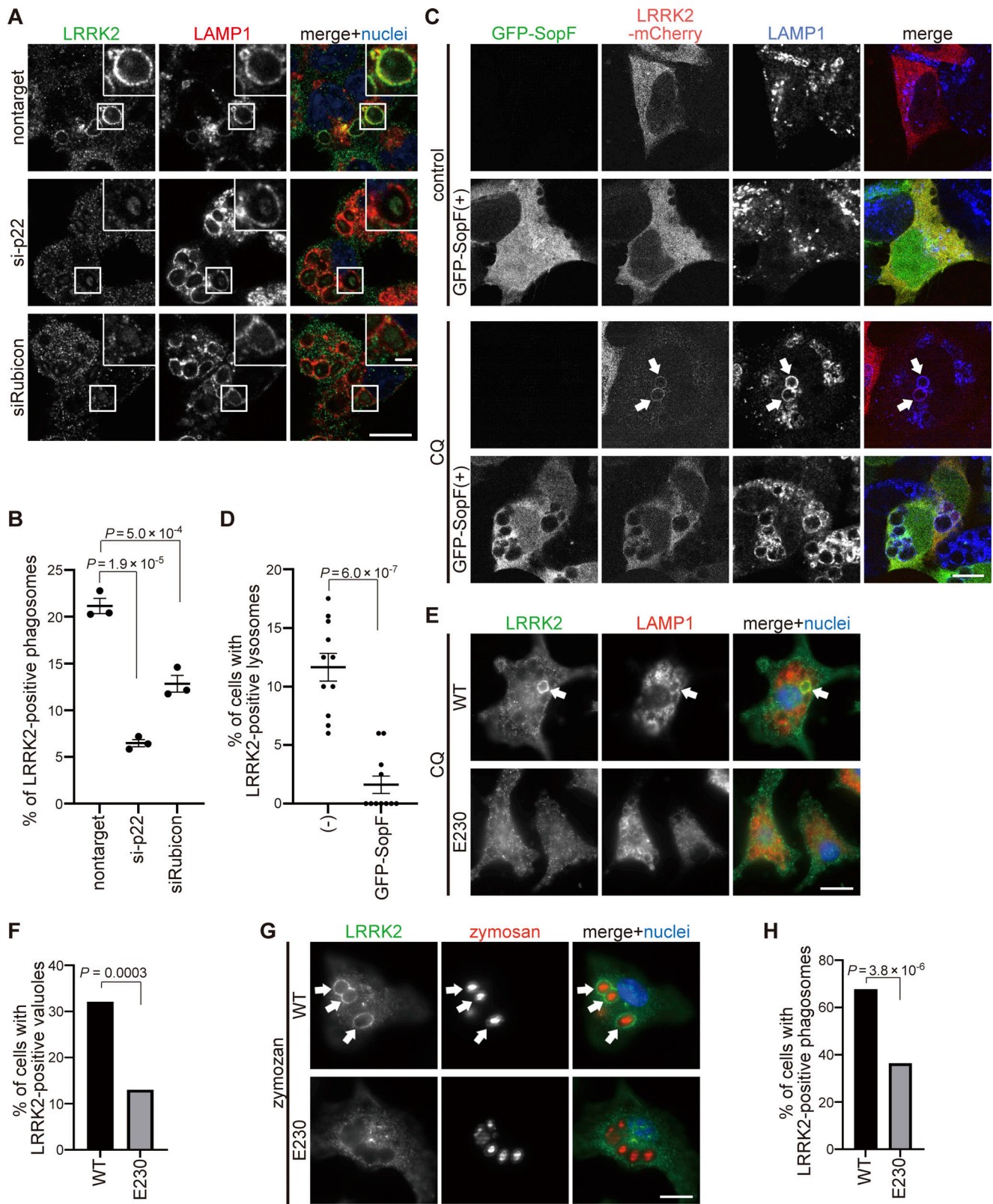

Figure 3. **Factors involved in LC3-assosiated phagocytosis and the V-ATPase–ATG16L1 axis are required for the targeting of LRRK2. (A)** Fluorescence images of LRRK2 in zymosan-treated RAW264.7 cells transfected with the indicated siRNAs. Scale bars, 10 and 1 μm (inset). **(B)** Percentage of LRRK2-positive phagosomes in total phagosomes, as shown in A. Data represent mean ± SEM ($N = 3$ independent experiments). The differences were analyzed using one-way ANOVA with Tukey's test. **(C)** Fluorescence images of LRRK2 in HEK293 cells transfected with GFP-SopF. Arrows indicate LRRK2-positive lysosomes. Scale bar, 10 μm. **(D)** Percentage of cells harboring LRRK2-positive lysosomes in CQ-treated HEK293 cells, as shown in C. Data represent mean ± SEM ($N = 11$ independent experiments). The difference was analyzed using an unpaired two-tailed $t$ test. **(E)** Fluorescence images of LRRK2 in CQ-treated BMDMs (WT or E230).

2017; Zhou et al., 2011) and Crohn's disease (Barrett et al., 2008; Hampe et al., 2007; Rioux et al., 2007), suggesting that LRRK2 and the ATG8 conjugation system components could act in a common pathway contributing to the pathogenesis of immune-related diseases in addition to PD. How they are involved in the pathomechanisms of these diseases should be further explored.

## Materials and methods
### Plasmids and siRNAs
The following plasmids were used in this study: pcDNA5 FRT/TO GFP-LRRK2 (Medical Research Council Protein Phosphorylation and Ubiquitylation Unit [MRC-PPU]), pcDNA5D frtTO mCherry-LRRK2 (MRC-PPU), 3×FLAG-human LRRK2 (Fujimoto

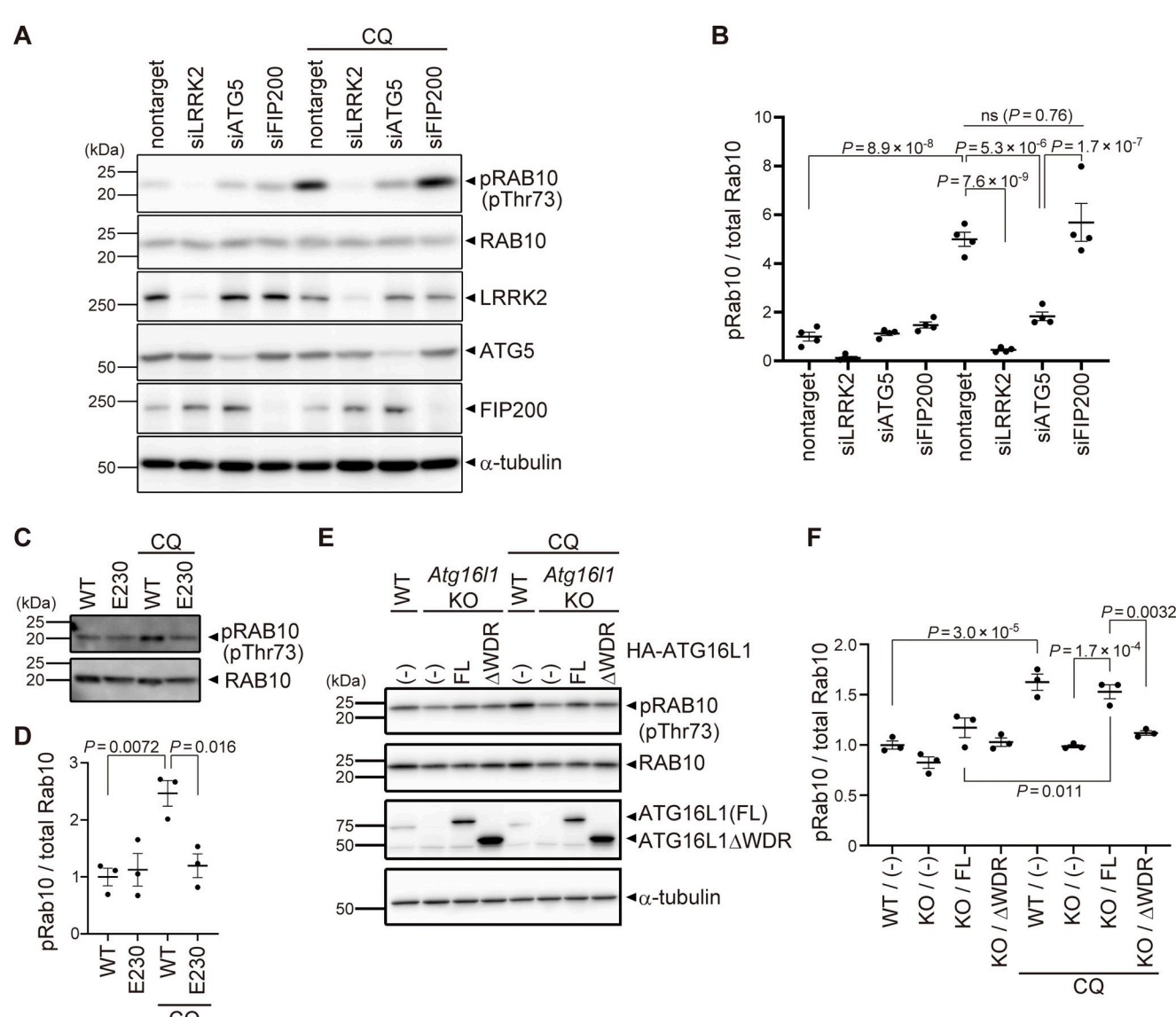

Figure 4. **Kinase activity of LRRK2 is regulated by the ATG8 conjugation system. (A)** Levels of phosphorylated Rab10 (pThr73) and other indicated proteins in CQ-treated or untreated RAW264.7 cells transfected with the indicated siRNAs. **(B)** Quantification of pRab10 divided by total Rab10 in RAW264.7 cells, as shown in A. Data represent mean ± SEM (*N* = 4 independent experiments). The difference was analyzed using one-way ANOVA with Tukey's test. **(C)** Levels of phosphorylated Rab10 (pThr73) in CQ-treated or untreated BMDMs derived from WT and E230 mice. **(D)** Quantification of pRab10 divided by total Rab10 in BMDMs, as shown in C. Data represent mean ± SEM (*N* = 4 independent experiments). The difference was analyzed using one-way ANOVA with Tukey's test. **(E)** Levels of pRab10 in CQ-treated or untreated MEF cells. *Atg16l1* KO MEF cells were infected with full-length (FL) or WD40 repeat-deficient (ΔWDR) ATG16L1. **(F)** Quantification of pRab10 divided by total Rab10 in MEF cells, as shown in E. Data represent mean ± SEM (*N* = 3 independent experiments). The difference was analyzed using one-way ANOVA with Tukey's test. Source data are available for this figure: SourceData F4.

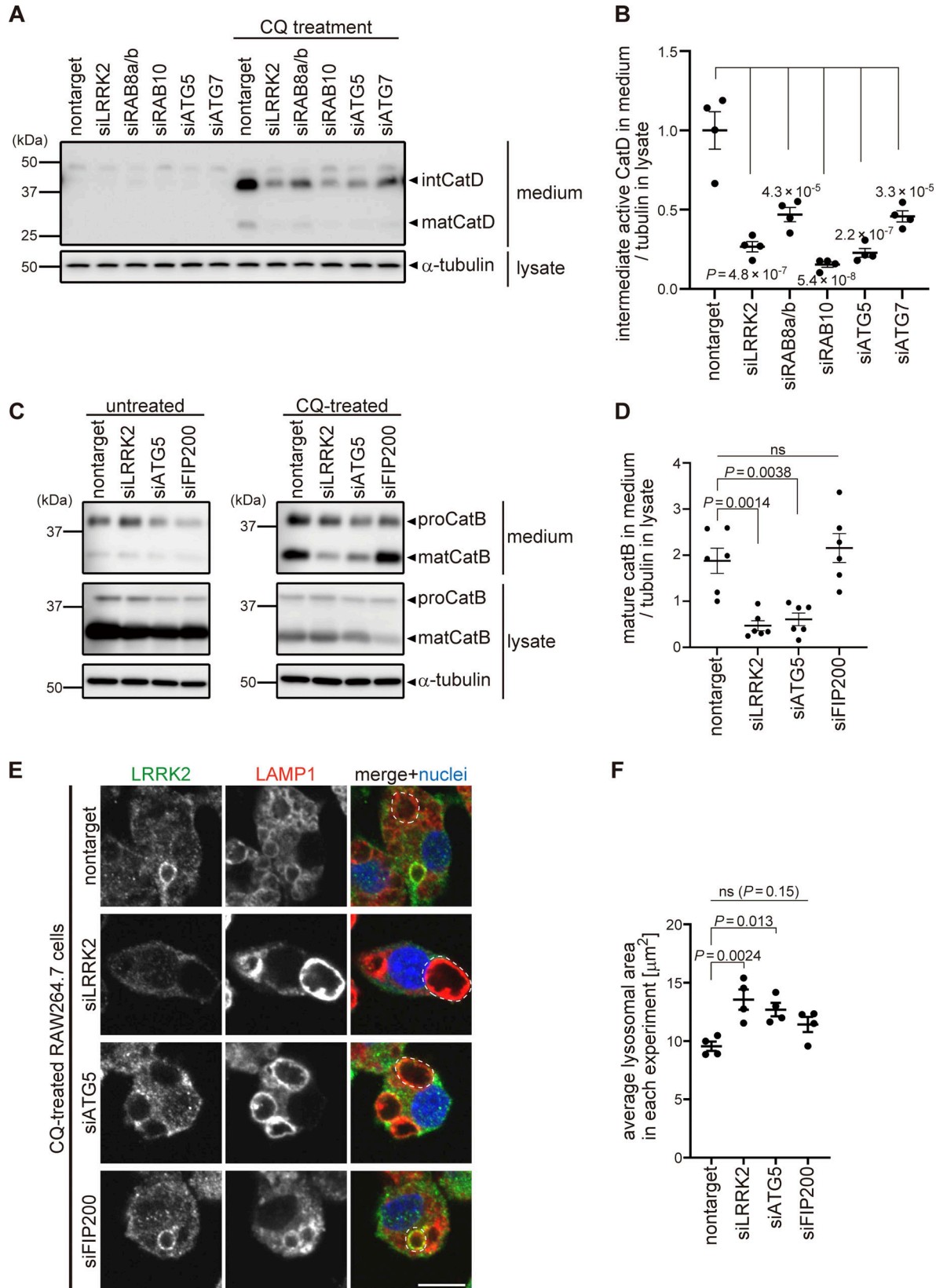

Figure 5. **The ATG8 conjugation system facilitates lysosome maintenance by LRRK2 independently of the autophagy initiation complex. (A)** Levels of mature (mat) and intermediate (int) CatD released into culture media from RAW264.7 cells transfected with the indicated siRNAs and treated with or without CQ. α-Tubulin in cell lysate was analyzed to normalize the amount of cells in each dish. **(B)** Quantification of CatD released into culture media upon CQ treatment, as shown in A. Data represent mean ± SEM ($N$ = 4 independent experiments). The difference was statistically analyzed using one-way ANOVA with

Tukey's test. P values compared to nontarget siRNA-transfected cells are shown in the graph. **(C)** Levels of mature (mat) CatB released into culture media from RAW264.7 cells transfected with the indicated siRNAs and treated with or without CQ. **(D)** Quantification of CatB released into culture media upon CQ treatment, as shown in C. Data represent mean ± SEM (N = 6 independent experiments). The difference was statistically analyzed using one-way ANOVA with Tukey's test. **(E)** Fluorescence images of morphologies of LAMP1-positive lysosomes in RAW264.7 cells transfected with the indicated siRNAs and treated with CQ. The largest lysosome in each cell is surrounded by a broken line. Scale bar, 10 μm. **(F)** Quantification of the size of the most enlarged lysosomes in each CQ-treated cells, as shown in E. The average size in each experiment was calculated and statistically analyzed. Data represent mean ± SEM (N = 4 independent experiments). The difference was analyzed using one-way ANOVA with Dunnett's test. Source data are available for this figure: SourceData F5.

---

et al., 2018), pEGFP-LC3, pEGFP-p62, 3×FLAG-LC3A/B/C (Chino et al., 2019), 3×FLAG-GABARAP/L1/L2 (Chino et al., 2019), and pEGFP-C1-SopF (#137734; Addgene). All siRNAs used in this study were purchased from Horizon Discovery (Dharmacon si-GENOME). Each siRNA contained a mixture of four different siRNAs per target (SMARTpool), except for siATG5, which contained a single siRNA (Cat No. D-064838-01). cDNA encoding human ATG16L1 (isoform β, NP_110430.5) with 3×HA tag added to the N-terminus was inserted into pMRX (Saitoh et al., 2002). To construct ATG16L1 lacking the WD40 domain, the Leu337stop mutation was introduced by site-directed mutagenesis. cDNA encoding human p62 (NP_003891.1) or WIPI2 (isoform b, NP_057087.2) with EGFP tag added to the N-terminus was inserted into pMRX.

### Antibodies and reagents
The following primary antibodies were used in this study: anti-LRRK2 (MJFF2 [c41-2]; Abcam), anti-human LAMP1 (D2D11; Cell Signaling Technology), anti-mouse LAMP1 (1D4B; Bio-Rad), anti-LC3 (PM036; MBL), anti-LC3 (L7543; Sigma-Aldrich), anti-GFP (A-11122; Invitrogen), anti-p62 (61832; BD Pharmingen), anti-galectin-3 (M3/38; BioLegend), anti-α-tubulin (DM1A; Sigma-Aldrich), anti-CatD (ERP3057Y; ab75852), anti-CatB (D1C7Y; Cell Signaling Technology), anti-Rab10 (D36C4; Cell Signaling Technology), anti-phospho-Thr73 Rab10 (MJF-R21 [ab230261]; Abcam), anti-ATG16L1 (D6D5; CST), anti-ATG5 (A0731; Sigma-Aldrich), anti-FIP200 (17250-1-AP; Proteintech), anti-ATG13 (SAB4200100; Sigma-Aldrich), and anti-Rab29 (MJF-R30-124; Abcam). The following reagents were used at final concentrations as indicated: CQ (50–100 μM; Sigma-Aldrich), LLOMe (1 mM; Sigma-Aldrich), and zymosan (Sigma-Aldrich).

### Cell culture and transfection
HEK293 cells and HEK293T cells were cultured in DMEM supplemented with 10% (vol/vol) FBS and in 100 U/ml penicillin and 100 μg/ml streptomycin at 37°C in a 5% $CO_2$ atmosphere. RAW264.7 cells and Lrrk2 KO RAW264.7 cells (American Type Culture Collection) were cultured on culture dishes for suspended cells (Sumitomo Bakelite Co.) under the same conditions as for the culture of HEK293 cells. WT and Atg16l1 KO MEF cells (Saitoh et al., 2008) were cultured under the same condition as for the culture of HEK293 cells. BMDMs were generated from femurs and tibias of WT and E230 mice as described previously (Rai et al., 2019; Wang et al., 2021). Briefly, macrophages were generated from adherent cells in RPMI-1640 containing 10% FCS and macrophage colony-stimulating factor (315-02; Peprotech) (30 ng/ml) for 6 d. Macrophage populations were quantified by FACS using antibodies against CD16/CD32, F4/80, and CD11b (101320, 123107; BioLegend). All experiments with mice were performed in accordance with UK Home Office guidelines and

under the UK Animals (Scientific Procedures) Act 1986. RAW264.7 cells and BMDMs were activated by IFN-γ treatment for 48 h before each assay. Transfection of plasmids and siRNA was performed using Lipofectamine LTX (Thermo Fisher Scientific) and Lipofectamine RNAiMAX (Thermo Fisher Scientific), respectively, according to the manufacturer's protocols. Cells were analyzed 48 h after siRNA transfection. For immunocytochemistry, cells were seeded on a coverslip.

### Preparation of retrovirus and generation of stable cell lines
To prepare retroviruses, HEK293T cells were transiently transfected with a retroviral vector together with pCG-VSV-G and pCG-gag-pol (gifts from Dr. T. Yasui, National Institutes of Biomedical Innovation, Health and Nutrition, Osaka, Japan) using Lipofectamine 2000 (Thermo Fisher Scientific). 3 d after transfection, culture supernatants were collected and filtrated using 0.22-μm membrane filter (Millipore). MEF cells were infected with retrovirus and selected with puromycin (Sigma-Aldrich).

### Immunocytochemistry
Immunofluorescence analysis was performed based on our previous methods (Eguchi et al., 2018; Sakurai and Kuwahara, 2021). Cells were fixed with 4% (wt/vol) paraformaldehyde for 20–30 min, washed with PBS, and then immersed in 100% EtOH at −20°C overnight. Cells were again washed with PBS and permeabilized and blocked with 3% (wt/vol) BSA in PBS containing 0.1% Triton X-100. Primary antibodies and corresponding secondary antibodies conjugated with Alexa Fluor dyes (Thermo Fisher Scientific) were diluted in the blocking buffer. The cells were incubated with primary antibody solutions for 2 h and then secondary antibody solutions for 1 h at room temperature. Nuclei were stained with DRAQ5 (1:3,000; Biostatus) or DAPI (1:5,000; Thermo Fisher Scientific) at the same time with secondary antibodies. The samples were imaged using a confocal laser scanning microscope system (FV3000 combined with IX83 inverted microscope; Evident), with a 100× oil-immersion objective lens (NA: 1.40, UPLSAPO100XO; Evident) and captured with FluoView FV31S-SW software (Evident). Image contrast and brightness were adjusted using a photo retouch software, GIMP2 (Spencer Kimball, Peter Mattis, and the GIMP Development Team). For the quantitative analysis of localization of LRRK2 on LAMP1-positive area, MCC on the fluorescence images was calculated using the ImageJ JACoP plugin, with the autothreshold method.

### CLEM
For observation of the LRRK2-LC3 double-positive structure, HEK293 cells expressing GFP-LC3 and LRRK2-mCherry were

grown for 2 d on a glass-bottom dish with 150-µm grids (TCI-3922-035R-1CS; Iwaki, a custom-made product based on 3922-035, with cover glass attached in the opposite direction) coated with carbon by vacuum evaporator (IB-29510VET; JEOL) and 0.1% gelatin. The cells were treated with CQ for 24 h before fixation. For observation of the GFP-LC3 and LRRK2 mCherry double-positive signal, the cells were fixed with freshly prepared 2% PFA with 0.5% glutaraldehyde in 0.1 M phosphate buffer (pH 7.4). After washing with 0.1 M phosphate buffer three times, cells were observed by the FV3000 confocal laser scanning microscope system (Evident) with a 60× oil-immersion objective lens (NA: 1.40, PLAPON60X; Evident) and then post-fixed overnight with 2.5% glutaraldehyde in 0.1 M sodium cacodylate buffer at 4°C. After washing with 0.1 M sodium cacodylate, cells were treated with 1% osmium tetroxide in 0.065 M sodium cacodylate buffer for 2 h at 4°C and rinsed five times using Milli-Q water. The samples were then stained with 2% uranyl acetate solution for 1 h and rinsed five times using Milli-Q water. The samples were dehydrated with ethanol series, covered with an EPON812-filled plastic capsule which invertedly stood on the sample surface, and polymerized at 40°C for 12 h and 60°C for 48 h. After polymerization, coverglasses were removed by soaking in liquid nitrogen and the sample block was trimmed to about 150 × 150 µm, retaining the same area where the fluorescent microscopic image was obtained. Then, serial sections (25 nm thick) were cut, collected on a silicon wafer strip, and then observed under a scanning electron microscope (JEM7900F; JEOL). CLEM images were constructed using FIJI (National Institutes of Health) and Photoshop CC software (Adobe).

### Immunoprecipitation

Cells were washed with PBS and lysed in a lysis buffer containing 50 mM Tris HCl, pH 7.6, 150 mM NaCl, 0.5% (vol/vol) Triton X-100, Complete EDTA-free protease inhibitor cocktail 3 (Roche), and PhosSTOP (Roche). Lysates were centrifuged at 20,800 × $g$ for 5 min at 4°C and supernatants were mixed with GFP-Trap beads (Chromotek) that had been equilibrated, followed by rotation for 2 h at 4°C. The samples were then centrifuged at 2,500 × $g$ to remove the unbound supernatant from the beads. The beads were washed with TBS buffer (50 mM Tris-HCl, pH = 7.6, 150 mM NaCl) five times, and were boiled in 2× sample buffer for 10 min at 90°C.

### Biochemical isolation of lysosomes

Isolation of lysosomes was conducted as described previously (Eguchi et al., 2018; Komori et al., 2023). Cells on a 10-cm dish were cultured in DMEM containing 1 mM HEPES-NaOH (pH 7.2) and 10% dextran-coated magnetite (DexoMAG 40; Liquids Research Ltd.) for 24 h, and then chased in normal media for 24 h. Cells were harvested with trypsin, centrifuged at 60 × $g$ for 5 min, washed with ice-cold PBS, lysed in 2 ml of ice-cold Buffer A (1 mM HEPES, 15 mM KCl, 1.5 mM Mg(Ac)$_2$, 1 mM DTT, protease/phosphatase inhibitors) with a Dounce homogenizer, and passed through a 23G needle for eight times. After homogenization, 500 µl of ice-cold Buffer B (220 mM HEPES, 375 mM KCl, 22.5 mM Mg(Ac)$_2$, 1 mM DTT, 0.1 mM sucrose, and 50 µg/

ml DNase I) was immediately added, and samples were inverted for five times, incubated for 5 min, and then centrifuged at 400 × $g$ for 10 min. The supernatant was then applied to an MS Column (Miltenyi Biotec) set on an OctoMACS separator (Miltenyi Biotec) and equilibrated with 0.5% BSA in PBS and the flow through was collected. 1 ml of DNase I solution (50 µg/ml DNase I, 0.1 mM sucrose in PBS) was added and the column was incubated for 10 min and washed with 1 ml of ice-cold sucrose buffer (0.1 mM sucrose in PBS). After removing the column from the OctoMACS separator, lysosomes were eluted with 250 µl of ice-cold sucrose buffer using a plunger.

### SDS-PAGE and immunoblot analysis

Immunoblot analysis was performed based on our previous methods (Eguchi et al., 2018; Sakurai and Kuwahara, 2021). Cells were washed with PBS on ice and lysed in a lysis buffer containing 50 mM Tris HCl, pH 7.6, 150 mM NaCl, 0.5% (vol/vol) Triton X-100, Complete EDTA-free protease inhibitor cocktail 3 (Roche), and PhosSTOP phosphatase inhibitor Cocktail (Roche). Lysates were centrifuged at 20,800 × $g$ for 5 min at 4°C and supernatants were mixed with NuPAGE LDS Sample Buffer (4 ×) buffer (Thermo Fisher Scientific). For SDS-PAGE, samples were loaded onto Tris-glycine gels and electrophoresed. After electrophoresis, samples were transferred to polyvinylidene fluoride membranes. Transferred membranes were blocked and incubated with primary antibodies and then with HRP-conjugated secondary antibodies (Jackson ImmunoResearch). Protein bands were detected by LAS-4000 (FUJIFILM). The integrated densities of protein bands were calculated using ImageJ software.

### Morphometric analysis of lysosomes

Lysosomes were fluorescence stained with an anti-LAMP1 antibody and images of cells were acquired using a confocal microscope. The area of the largest lysosome in each cell was measured using ImageJ software (National Institutes of Health) and shown in dot plots. The average size of the largest lysosomes in each condition was calculated and mean values of the average from three to five independent experiments are shown in bar graphs. A total of cells on a coverslip were analyzed for each condition in each experiment.

### Measurement of CatB/D in media

IFN-γ–activated RAW264.7 cells were cultured in DMEM containing 1% FBS for 3 h. Some cells were cultured in the presence of CQ (100 µM) for 3 h. Media were collected and centrifuged at 200 × $g$ for 5 min. Cells were lysed and analyzed by immunoblotting. Supernatants of media were analyzed by immunoblotting. For immunoblotting, the supernatants were mixed with NuPAGE LDS Sample Buffer (4×) buffer (Thermo Fisher Scientific).

### Statistics

The statistical significance of the difference in mean values was calculated by unpaired, two-tailed Student's $t$ test or one-way ANOVA using GraphPad Prism (9.4.0). The statistical significance of the difference of the percentages between two groups

was calculated by Fisher's exact test. P values <0.05 were considered statistically significant. No exclusion criteria were applied to exclude samples or animals from analysis.

## Online supplemental material

Fig. S1 shows the immunostaining of LRRK2 knockout cells and two more CLEM images of LC3 and LRRK2 on lysosomal single membranes, in addition to Fig. 1 F images. Fig. S2 shows the assessment of ATG5-dependent recruitment of LRRK2 by MCC measurement and biochemical analysis. Fig. S3 shows knockdown effects of ATG13 and FIP200. Fig. S4 shows a lack of binding between LRRK2 and ATG8 family proteins and no role of the ATG8 conjugation system on Rab29 recruitment. Fig. S5 shows a morphological analysis of lysosomes under siRNA treatment and without CQ.

## Data availability

All data sets used or analyzed in this study are available from the corresponding author upon request.

## Acknowledgments

The authors thank Mr. Shoji Yamaoka, The University of Tokyo, Tokyo, Japan, for providing pMRX-IP, Mr. Teruhito Yasui, The University of Tokyo, Tokyo, Japan, for pCG-VSV-G and pCG-gag-pol, Ms. Yoko Ishida and Ms. Keiko Igarashi, The University of Tokyo, Tokyo, Japan, for their technical support on CLEM experiment and analysis, and the Iwatsubo lab members for helpful suggestions and discussions.

This work was supported by the Japan Society for the Promotion of Science KAKENHI grant numbers 16K07039 (T. Kuwahara), 19K07816 (T. Kuwahara), 22H02949 (T. Kuwahara), 22H04638 (T. Kuwahara), 20H00525 (T. Iwatsubo), 19K16118 (T. Eguchi), 22K15057 (T. Eguchi), 21J12881 (M. Sakurai), The University of Tokyo World-leading INnovative Graduate Study Program for Life Science and Technology (WINGS-LST) collaborative research grant (M. Sakurai), and by the Exploratory Research for Advanced Technology research funding program of the Japan Science and Technology Agency (JPMJER1702) (N. Mizushima). Work by T. Wileman and Y. Wang was funded in part by through Biology and Biotechnology Research Council grant BB/W002450/1. Open Access funding provided by University of Tokyo.

Author contributions: Conceptualization: T. Eguchi, M. Sakurai, and T. Kuwahara; Investigation: T. Eguchi, M. Sakurai, Y. Wang, C. Saito, G. Yoshii, and T. Kuwahara; Methodology: T. Eguchi, M. Sakurai, and C. Saito; Resources: T. Wileman; Data curation: T. Eguchi, M. Sakurai, and T. Kuwahara; Writing—original draft: T. Eguchi and T. Kuwahara; Writing—review and editing: M. Sakurai, C. Saito, T. Wileman, N. Mizushima, and T. Iwatsubo; Supervision: T. Kuwahara and T. Iwatsubo; Project administration: T. Kuwahara; Funding acquisition: T. Eguchi, M. Sakurai, T. Wileman, N. Mizushima, T. Kuwahara, and T. Iwatsubo. All authors read and approved the final manuscript.

Disclosures: C. Saito reported personal fees from JEOL outside the submitted work; and "lectured on CLEM at a JEOL webinar (in Japanese) and received an honorarium for the presentation." No other disclosures were reported.

Submitted: 16 February 2023

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

# Supplemental material

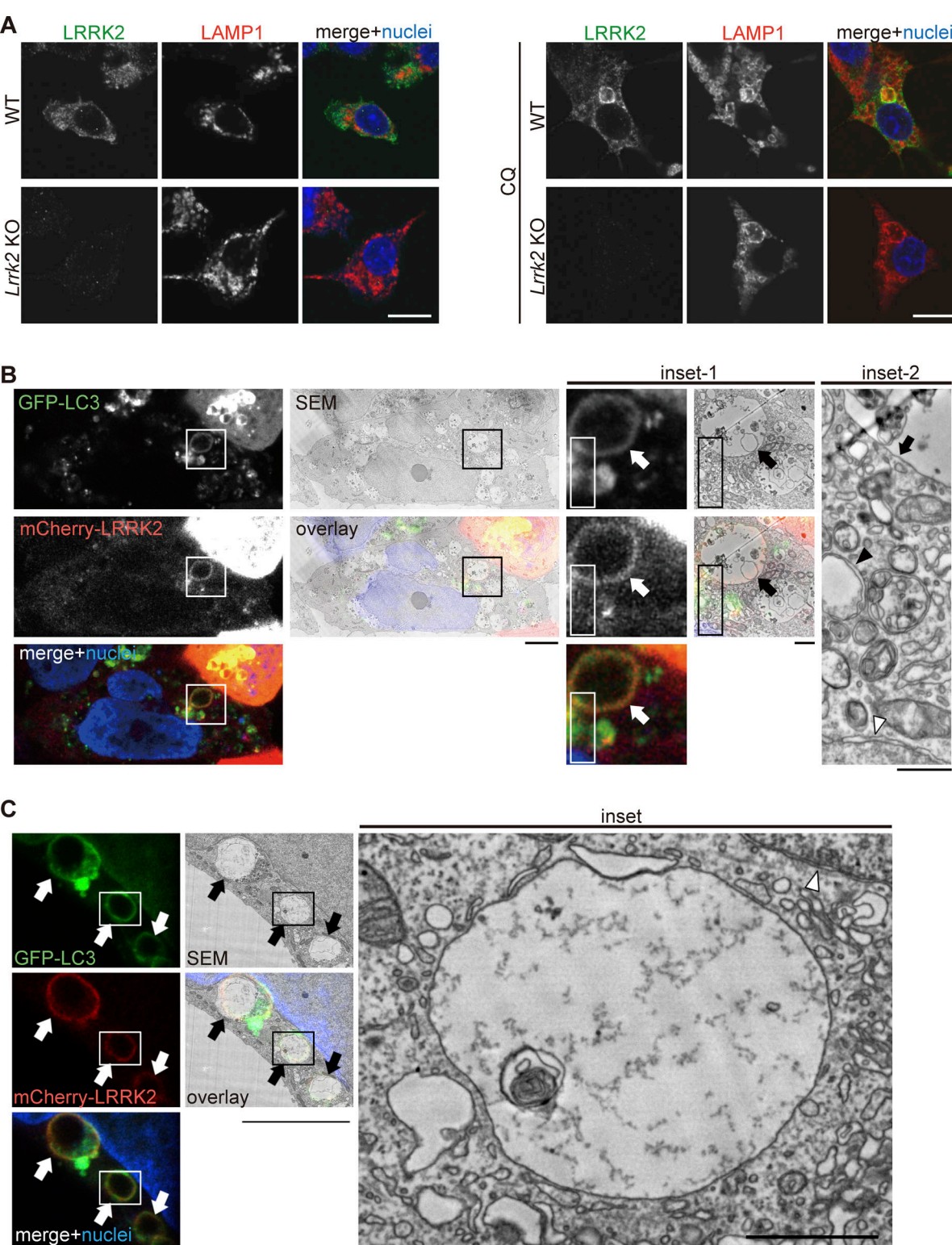

Figure S1. **Detailed analyses on LRRK2 localization. (A)** Confirmation of the specificity of LRRK2 immunostaining. Immunofluorescence signal of LRRK2 stained with MJFF2 antibody was not detected in *Lrrk2* KO RAW264.7 cells in the presence or absence of CQ treatment. Scale bars, 10 μm. **(B and C)** Two representative CLEM images, in addition to Fig. 1 F, showing colocalization of GFP-LC3 and mCherry-LRRK2 on lysosomal single membranes under CQ treatment. Scale bars, 5 μm (B), 10 μm (C), 1 μm (inset-1 in B and inset in C), 500 nm (inset-2 in B). Arrows: LRRK2-LC3 double-positive membranes, black arrowhead in B: autolysosome, white arrowheads: nuclear envelope.

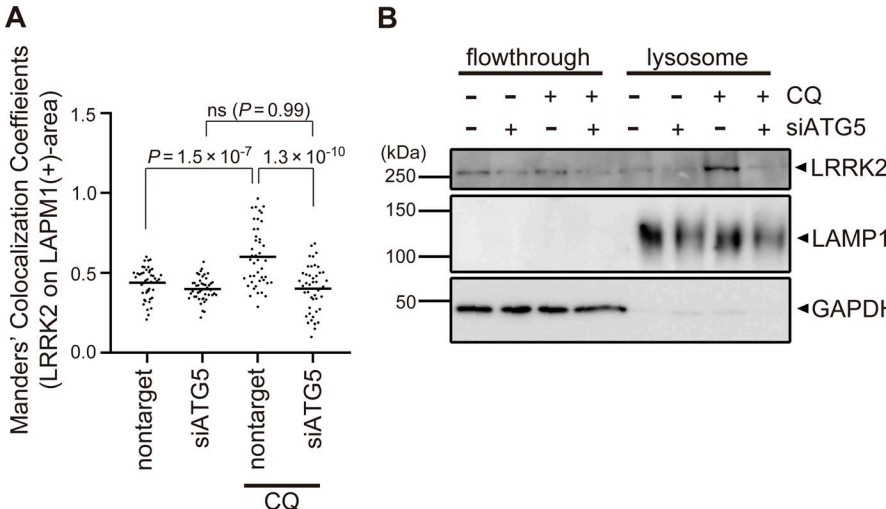

Figure S2.  **Confirmation of ATG5-dependent recruitment of LRRK2 to lysosomes upon CQ treatment. (A)** MCC analysis showing the ratio of LRRK2 on LAMP1-positive area in CQ-treated and untreated RAW264.7 cells. The difference was analyzed using one-way ANOVA with Tukey's test. **(B)** Biochemical isolation of lysosomes from RAW264.7 cells showing the enrichment of LRRK2 in lysosomal fraction upon treatment with CQ but not with siATG5. Source data are available for this figure: SourceData FS2.

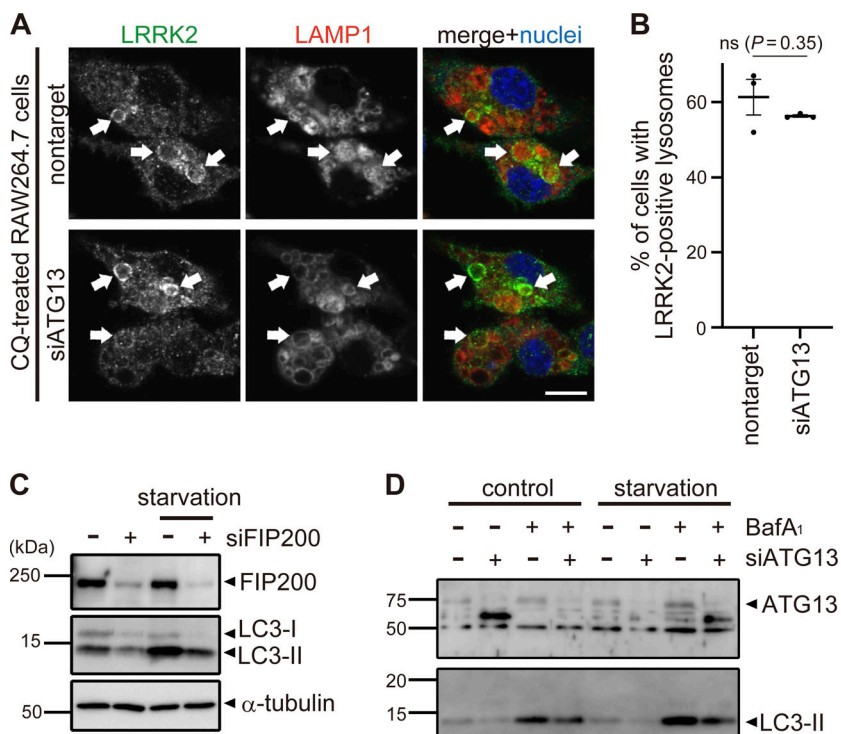

Figure S3.  **Analyses of knockdown effects of ATG13 and FIP200. (A)** Fluorescence images of endogenous LRRK2 in RAW264.7 cells transfected with nontarget or ATG13 siRNA and treated with CQ. Arrows indicate LRRK2-positive lysosomes. Scale bar, 10 μm. **(B)** Percentages of cells harboring LRRK2-positive lysosomes as shown in A. Data represent mean ± SEM (N = 3 independent experiments). The difference was analyzed using one-way ANOVA with Dunnet's test. ns, not significant. **(C and D)** Immunoblot pictures showing efficient knockdown of FIP200 (C) or ATG13 (D) as well as resultant decrease of starvation-induced autophagy as assessed by LC3 lipidation in RAW264.7 cells treated with siFIP200 or siATG13. Source data are available for this figure: SourceData FS3.

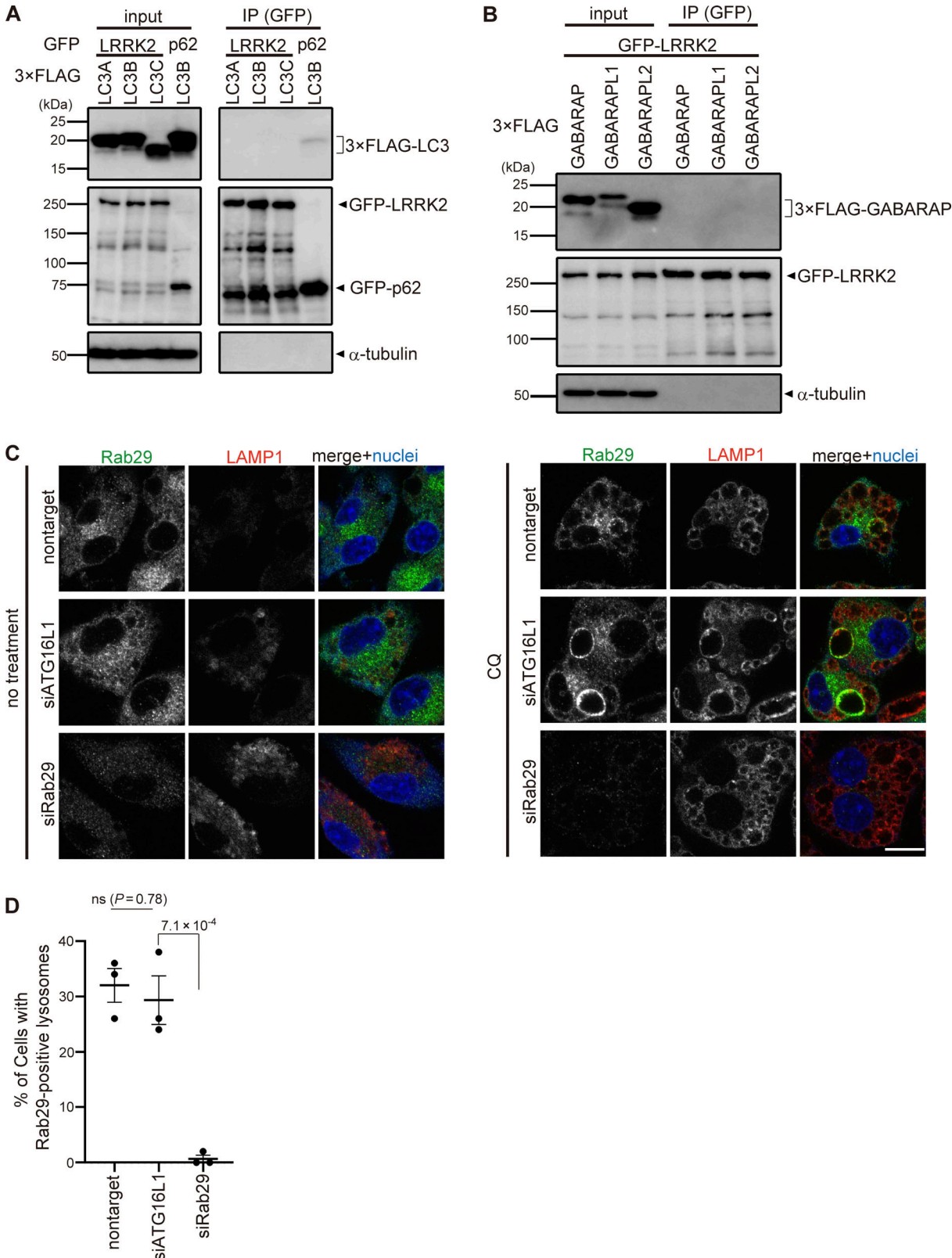

Figure S4. **Analyses on the possibility of LRRK2 recruitment via ATG8 or Rab29. (A and B)** Lack of binding between LRRK2 and ATG8 family proteins. Immunoprecipitation (IP) analysis of HEK293 cells overexpressing 3×FLAG-taggted LC3A/B/C (A) or GABARAP/GABARAPL1/GABARAPL2 (B) together with GFP-tagged LRRK2 or p62. Pulldown using GFP-trap resulted in the precipitation of LC3 with p62 but not with LRRK2. **(C)** Fluorescence images of endogenous Rab29 in RAW264.7 cells transfected with siATG16L1 or siRab29 (positive control) and treated with (right) or without (left) CQ. Scale bar, 10 μm. **(D)** Percentage of cells harboring Rab29-positive lysosomes, as shown in C. Data represent mean ± SEM (N = 3 independent experiments). The difference was analyzed using one-way ANOVA with Dunnet's test. Source data are available for this figure: SourceData FS4.

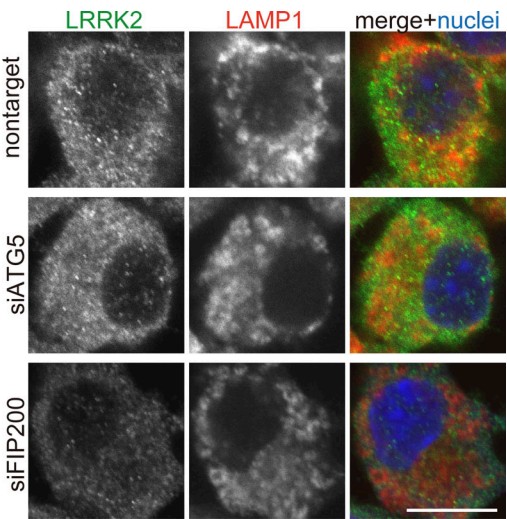

Figure S5.   **Morphological analysis of lysosomes in the absence of CQ.** Representative fluorescence images of LAMP1-positive lysosomes as well as LRRK2 in RAW264.7 cells transfected with the indicated siRNAs, without following CQ treatment. Scale bar, 10 μm.

