## [Peer Review File · The Journal of Cell Biology]

The V-ATPase-ATG16L1 axis recruits LRRK2 to facilitate the lysosomal stress response

Tomoya Eguchi, Maria Sakurai, Yingxue Wang, Chieko Saito, Gen Yoshii, Thomas Wileman, Noboru Mizushima, Tomoki Kuwahara, and Takeshi Iwatsubo

Corresponding Author(s): Tomoki Kuwahara, The University of Tokyo

Review Timeline:

Submission Date:	2023-02-16
Editorial Decision:	2023-03-24
Revision Received:	2023-10-13
Editorial Decision:	2023-11-28
Revision Received:	2023-12-04
Accepted:	2023-12-11

Monitoring Editor: Hong Zhang

Scientific Editor: Tim Fessenden

Transaction Report:

DOI: <https://doi.org/10.1083/jcb.202302067>

March 24, 2023

Re: JCB manuscript #202302067

Dr. Tomoki Kuwahara
The University of Tokyo
Dept. Neuropathology, Grad. Sch. Med.
7-3-1, Hongo, Bunkyo-ku
Tokyo 1130033
Japan

Dear Dr. Kuwahara,

Thank you for submitting your manuscript entitled "Non-autophagic ATG8 conjugation to stressed lysosomes recruits LRRK2 to facilitate stress responses". Your manuscript has been assessed by expert reviewers, whose comments are appended below. Although the reviewers express potential interest in this work, significant concerns unfortunately preclude publication of the current version of the manuscript in JCB.

As you will see, reviewers agreed that the novel roles observed for ATG8 family proteins with LRRK2 are conceptually interesting. However, all felt that this work is missing important mechanistic details as well as controls whose inclusion would clarify the authors' model. In particular, reviewers sought details on ATG8, ATG5/ATG16L, and LRRK2. Multiple reviewers also requested confirmation of siRNA knockdowns, and asked for clarification of data shown in Figure 5. Reviewer 1 also raised concerns with imaging quality, and Reviewer 2 sought additional data concerning autophagy initiation via Vps34-Beclin-1. While all reviewer requests should be addressed in some manner, additional data beyond those noted here are not required in a revision. Last, we felt that a manuscript revised along these lines would be appropriate in the Article format rather than the Report format.

Please let us know if you are able to address the major issues outlined above and wish to submit a revised manuscript to JCB, and if so please send us a revision plan. Note that a substantial amount of additional experimental data likely would be needed to satisfactorily address the concerns of the reviewers. The typical timeframe for revisions is three to four months. While most universities and institutes have reopened labs and allowed researchers to begin working at nearly pre-pandemic levels, we at JCB realize that the lingering effects of the COVID-19 pandemic may still be impacting some aspects of your work, including the acquisition of equipment and reagents. Therefore, if you anticipate any difficulties in meeting this aforementioned revision time limit, please contact us and we can work with you to find an appropriate time frame for resubmission. Please note that papers are generally considered through only one revision cycle, so any revised manuscript will likely be either accepted or rejected.

If you choose to revise and resubmit your manuscript, please also attend to the following editorial points. Please direct any editorial questions to the journal office.

General Guidelines for Articles:

Text limits: Character count is < 40,000, not including spaces. Count includes title page, abstract, introduction, results, discussion, and acknowledgments. Count does not include materials and methods, figure legends, references, tables, or supplemental legends.

Figures: Your manuscript may have up to 10 main text figures. To avoid delays in production, figures must be prepared according to the policies outlined in our Instructions to Authors, under Data Presentation, <https://jcb.rupress.org/site/misc/ifora.xhtml>. All figures in accepted manuscripts will be screened prior to publication.

*****IMPORTANT:** It is JCB policy that if requested, original data images must be made available. Failure to provide original images upon request will result in unavoidable delays in publication. Please ensure that you have access to all original microscopy and blot data images before submitting your revision. ***

Supplemental information: There are strict limits on the allowable amount of supplemental data. Your manuscript may have up to 5 supplemental figures. Up to 10 supplemental videos or flash animations are allowed. A summary of all supplemental material should appear at the end of the Materials and methods section.

Please note that JCB now requires authors to submit Source Data used to generate figures containing gels and Western blots with all revised manuscripts. This Source Data consists of fully uncropped and unprocessed images for each gel/blot displayed in the main and supplemental figures. Since your paper includes cropped gel and/or blot images, please be sure to provide one Source Data file for each figure that contains gels and/or blots along with your revised manuscript files. File names for Source Data figures should be alphanumeric without any spaces or special characters (i.e., SourceDataF#, where F# refers to the

associated main figure number or SourceDataFS# for those associated with Supplementary figures). The lanes of the gels/blots should be labeled as they are in the associated figure, the place where cropping was applied should be marked (with a box), and molecular weight/size standards should be labeled wherever possible.

If you choose to resubmit, please include a cover letter addressing the reviewers' comments point by point. Please also highlight all changes in the text of the manuscript.

Regardless of how you choose to proceed, we hope that the comments below will prove constructive as your work progresses. We would be happy to discuss them further once you've had a chance to consider the points raised. You can contact the journal office with any questions, cellbio@rockefeller.edu or call (212) 327-8588.

Thank you for thinking of JCB as an appropriate place to publish your work.

Sincerely,

Hong Zhang
Monitoring Editor
Journal of Cell Biology

Tim Fessenden
Scientific Editor
Journal of Cell Biology

Reviewer #1 (Comments to the Authors (Required)):

Non-autophagic ATG8 conjugation to stressed lysosomes recruits LRRK2 to facilitate stress responses by Eguchi, Sakurai et al

Eguchi, Sakurai and colleagues outline a series of cell biological experiments to evaluate mechanism(s) by which LRRK2 can be recruited to various membranes in the endolysosomal system after cell stress, which this group and several others have now confirmed is a robust phenomenon. Experiments are presented to show that LRRK2 is recruited to LC3 positive lysosomes/phagosomes (figure 1) and that recruitment is blocked by siRNA against core autophagy machinery (figure 2) or LC3-based phagocytosis (figure 3). There are more modest effects of similar ATG manipulation on LRRK2 kinase activity as measured by pRab10 (figure 4) and some data suggesting that cathepsin secretion into the media is LRRK2 dependent but independent of ATG conjugation (Figure 5). Overall, the theoretical model is interesting but some aspects of mechanism are not well resolved and for a more conceptually impactful paper it would be important to develop several aspects. Specific points;

1. Throughout, I would like to see better resolution in the microscopy. For example, in figure 1A-C the LAMP1 staining is diffuse rather than punctate. This is particularly important for interpretation of recruitment where larger LAMP1+ structures are better resolved vs smaller ones - is there recruitment of LC3/LRRK2 to smaller LAMP1+ presumptive lysosomes? Rather than counting, it would be important to use Mander's coefficient.

2. In the same figure and later in the paper, the endogenous LRRk2 staining is based on the rabbit monoclonal antibody MJFF2 which is notorious for showing background staining. Given this is in RAW 264.7 cells and LRRK2 knockout RAW cells have been developed (<https://www.atcc.org/products/sc-6004>) it is required that this data is validated by showing lack of staining in knockouts.

3. I actually agree that figure 1D,E demonstrate that LRRK2 is not recruited to Galectin-3 positive heavily damaged lysosomes but the comparison between LLOMe and chloroquine is a bit misleading as the time course of lysosomal damage and LRRK2 activation is very different between these compounds. Authors should use both compounds at different times to show that LRRK2 activation occurs prior to establishment of Gal3+ lysosomes destined for lysophagy.

4. The CLEM in 1F is also not very convincing, in part because the contrast on the EM portions is very low making it hard to discern organelles and in part because the LRRK2 signal (here with mCherry, which reinforces the concern about endogenous staining) is clearly overexposed, presumably due to that cell having very high levels of transfected protein. I really think better images should be provided, ideally with more comprehensive sampling so we can evaluate the distinction between LRRK2 at lysosomes vs phagosomes.

5. Because of some of the concerns above about staining for LRRK2, and to be able to show robust data, figure 2 recruitment data needs to be validated by at least one orthogonal technique. Lyso-IP or subcellular fractionation would be appropriate.

6. For each siRNA in figure 2, please include validation that each has the expected knockdown at protein level, particularly for the negative result of FIP200. It would be helpful to show that autophagy initiation complex is rendered non-functional as a result.
7. The major concern with the manuscript as submitted is that the mechanism(s) involved are poorly developed. The interpretation of figures 2 and 3 is that LRRK2 recruitment to stressed lysosomes requires ATG8 conjugation but not the autophagy initiation complex, perhaps instead involving interaction with ATG18L1. Does LRRK2 physically interact with ATG18L1 under conditions of lysosomal stress? The discussion mentions data not presented suggesting not but if this is not true, then how is the ATG8 conjugation machinery involved in LRRK2 recruitment?
8. Figure 4A,B are confusing in the context of the rest of the data. Previously, in figure 2E it appeared that siRNA against ATG5 decreased LRRK2 recruitment to lysosomes substantially, from about 60% to 20% of cells. However, in figure 4B the same treatment only has an ~20% decrease in pRAB10 which is statistically significant but clearly much smaller than recruitment. Is LRRK2 still kinase active but away from the lysosome?
9. A similar concern for 4C,D; if understood correctly this data is from Atg16l1 knockout MEFs but this makes it hard to compare to the data in figure 3E-H which is from Atg16l1 'E320' BMDMs. Both assays should be performed in both genetic models and cell types.
10. I am honestly not sure what figure 5A-D tells us. While both knockdown of LRRK2 and ATG5 lowers cathepsin D and B in the media, this is not sufficient to infer that ATG5 affects LRRK2 as the effects could be independent and/or mechanistically unrelated. Evaluation of additive effects of ATG5 and LRRK2 should be performed and mechanism should be developed sufficiently to show that other aspects of Cathepsin biology (synthesis, trafficking, maturation) remain unaffected by both ATG5 and LRRK2 knockdown.
11. Figure 5E,F is similarly uncertain - there is a non-significant effect of ATG5 knockdown on lysosomal area but the importance of this is obscure. The experiment also does not include siRNA treatments without chloroquine, again raising the concern that these are independent (but not significant) effects.
12. RAB29 is invoked as a possible regulator in the discussion but this seems unlikely given that LRRK2 is still activated in RAB29 knockout cells (PMID: 33135724). Given the lack of mechanism in the paper, at least this is a testable idea that should be evaluated experimentally.

Reviewer #2 (Comments to the Authors (Required)):

The manuscript by Eguchi and colleagues reports a finding of the ATG8 ubiquitin-like conjugation system that controls the recruitment of Parkinson's disease-related protein LRRK2 and autophagy adaptor LC3 onto single-membrane of stressed lysosome or phagosomes. The recruitment of LRRK2 does not appear to require autophagy initiation regulators (e.g. FIP200). It may involve Atg16l-WD40 domain and perhaps v-ATPase activity. Furthermore, it shows that the ATG8 conjugation regulates LRRK2 activity (as measured with rab10 substrate phosphorylation) and lysosome enzyme release. Despite the interesting findings, the study could benefit from providing molecular mechanism whereby ATG8 conjugation regulates LRRK2 recruitment to stressed lysosomes. The following are my additional comments:

1. The observation seems to suggest that lipidated LC3 (II) may be the link for LRRK2 recruitment to the single membrane of stressed lysosomes. It also remains possible that other ATG8 family member (e.g. GABARAP) could be potential adaptors for the recruitment. It could be also interesting to test if LRRK2 contains conserved LIR, which may mediate the interaction between LRRK2 and ATG8 family members. Additional data from the above study would readily provide important insight and significantly improve the manuscript.
2. Related to the above point, change of LC3 lipidation status with Atg4 manipulation could be used to test their hypothesis for the purpose of strengthening their conclusion.
3. The author concluded that ATG8 conjugation but not autophagy initiation regulates the LRRK2 and LC3 recruitment. But the claim of "autophagy initiation" is based solely on the assay of FIP200. The study could also include the examination of the component in another autophagy initiation/nucleation regulator complex (Vps34-beclin1-Atg14) to back up the claim.
4. The manuscript should include the western blot analysis of various protein levels through siRNA knock-down to demonstrate the reduction of the proteins in their assays. This seems to be a problem for multiple figures.
5. The number of biological repeats shown in figure 2E is inconsistent with the figure legend.
6. The author should include LRRK2 inhibitor to show the specificity of rab10-phosphorylation antibody (figure 4)
7. Figure 5E shows LRRK2 recruitment to Lamp1+ vesicle in siATG5. This contradicts the previous images of in Figure 2C. Figure 5E also gives the impression that LRRK2- lysosomes are more enlarged than LRRK2+. It would be good to quantify and supplement the idea of LRRK2's importance in lysosome maintenance, especially considering the quantitation in Figure 5F has non-significant change.

Reviewer #3 (Comments to the Authors (Required)):

In the manuscript entitled "Non-autophagic ATG8 conjugation to stressed lysosomes recruits LRRK2 to facilitate stress responses", the authors explored the activation mechanism of LRRK2. They found LRRK2 is recruited to single membranes after lysosomal stress. This recruitment is independent of canonical autophagy machinery but relies on p22, Rubicon, and the V-

ATPase-ATG16L1 axis. They further demonstrate that these factors are required for the function of LRRK2, which is involved in maintaining lysosomal homeostasis. Through a series of experiments using microscopic and genetic techniques, they provided insights into the role of the V-ATPase-ATG16L1 axis in activating LRRK2. Overall, the manuscript is well-presented, and the figures are organized. However, in its current form, the manuscript is relatively simplified and has several questions that need to be addressed.

1. The title "Non-autophagic ATG8 conjugation to stressed lysosomes recruits LRRK2 to facilitate stress responses" is misleading. In fact, the authors did not directly demonstrate the activation of LRRK2 by non-autophagic ATG8 conjugation. It is possible that the ATG16L1 complex, activated by V-ATPase, can activate LRRK2 directly. Therefore, the authors must examine the LRRK2 phenotype in ATG8-deficient cells, otherwise the title should be changed to "the V-ATPase-ATG16L1 axis".
2. LAP (p22/Rubicon-dependent) and CASM (V-ATPase/ATG16L1 WD40-dependent and SopF sensitive) are two known non-canonical LC3 activation pathways. In this manuscript, it is surprising that both pathways are required for LRRK2 activation. The authors should explain why activation of LRRK2 requires both LAP and CASM, and what is the connection between these two pathways?
3. In the discussion, the authors mentioned the possibility that Rab29 is responsible for the regulation of LRRK2. Did the authors validate the role of Rab29 in their experimental system?
4. Besides the release of lysosomal cathepsins, are there other assays that indicate lysosomal homeostasis that could corroborate LRRK2 function?
5. The phenotype that ATG5/ATG16L1 regulates the kinase activity of LRRK2 is relatively weak in Fig. 4. The authors could emphasize this conclusion by detection of the level of phosphorylated Rab8, and by overexpression of SopF instead of ATG5 knockdown.

Author response to reviewers' comments

Editor's comment

Thank you for submitting your manuscript entitled "Non-autophagic ATG8 conjugation to stressed lysosomes recruits LRRK2 to facilitate stress responses". Your manuscript has been assessed by expert reviewers, whose comments are appended below. Although the reviewers express potential interest in this work, significant concerns unfortunately preclude publication of the current version of the manuscript in JCB.

As you will see, reviewers agreed that the novel roles observed for ATG8 family proteins with LRRK2 are conceptually interesting. However, all felt that this work is missing important mechanistic details as well as controls whose inclusion would clarify the authors' model. In particular, reviewers sought details on ATG8, ATG5/ATG16L, and LRRK2. Multiple reviewers also requested confirmation of siRNA knockdowns, and asked for clarification of data shown in Figure 5. Reviewer 1 also raised concerns with imaging quality, and Reviewer 2 sought additional data concerning autophagy initiation via Vps34-Beclin-1. While all reviewer requests should be addressed in some manner, additional data beyond those noted here are not required in a revision. Last, we felt that a manuscript revised along these lines would be appropriate in the Article format rather than the Report format.

>> We thank the editor for understanding the significance of our study and for raising important points that need to be addressed. We now believe we have addressed the reviewers' concerns as best as we can at this time. We have also reformatted our manuscript into Article type.

Reviewer #1 (Comments to the Authors (Required)):

Non-autophagic ATG8 conjugation to stressed lysosomes recruits LRRK2 to facilitate stress responses by Eguchi, Sakurai et al

Eguchi, Sakurai and colleagues outline a series of cell biological experiments to evaluate mechanism(s) by which LRRK2 can be recruited to various membranes in the endolysosomal system after cell stress, which this group and several others have now confirmed is a robust phenomenon. Experiments are presented to show that LRRK2 is recruited to LC3 positive lysosomes/phagosomes (figure 1) and that recruitment is blocked by siRNA against core autophagy machinery (figure 2) or LC3-based phagocytosis (figure 3). There are more modest effects of similar ATG manipulation on LRRK2 kinase activity as measured by pRab10 (figure 4) and some data suggesting that cathepsin secretion into the media is LRRK2 dependent but independent of ATG

conjugation (Figure 5). Overall, the theoretical model is interesting but some aspects of mechanism are not well resolved and for a more conceptually impactful paper it would be important to develop several aspects.

>> Thank you for understanding our study and its significance. As to the issue of modest effect of ATG manipulation on LRRK2 kinase activity in Fig 4A-B, we now confirmed a higher effect by using a different siRNA for ATG5. The detail is explained in the 8th point below.

Specific points;

1. Throughout, I would like to see better resolution in the microscopy. For example, in figure 1A-C the LAMP1 staining is diffuse rather than punctate. This is particularly important for interpretation of recruitment where larger LAMP1+ structures are better resolved vs smaller ones - is there recruitment of LC3/LRRK2 to smaller LAMP1+ presumptive lysosomes? Rather than counting, it would be important to use Mander's coefficient.

>> We have changed all pictures in Fig 1A-C to those with better resolution using new confocal microscope and displayed them slightly larger. LC3 and LRRK2 were mostly localized to large LAMP1+ vesicles upon CQ or zymosan treatment (Fig 1A), whereas LC3 (but not LRRK2) was found on smaller LAMP1+ structures upon starvation (Fig 1C).

We have also analyzed the colocalization of LRRK2 and LAMP1 using Manders' colocalization coefficient (MCC) and the results were shown in new Fig S2A, which includes the conditions under ATG5 knockdown. As expected, the MCC between LRRK2 and LAMP1 signals was significantly increased upon CQ exposure in nontarget siRNA-treated cells and not in siATG5-treated cells. The sensitivity of detecting LRRK2 recruitment using MCC was lower than that by the counting method probably due to the incorporation of weak background staining.

2. In the same figure and later in the paper, the endogenous LRRK2 staining is based on the rabbit monoclonal antibody MJFF2 which is notorious for showing background staining. Given this is in RAW 264.7 cells and LRRK2 knockout RAW cells have been developed (<https://www.atcc.org/products/sc-6004>) it is required that this data is validated by showing lack of staining in knockouts.

>> We have obtained LRRK2 KO RAW264.7 cells from ATCC and immunostained for endogenous LRRK2 using MJFF2 antibody. The results are shown in new Fig S1A; we could confirm the lack of staining in LRRK2 KO cells. Please note that we previously found that clear immunostaining with MJFF2 needs additional procedure, i.e., ethanol postfixation/delipidation after PFA fixation, as described in Eguchi et al, *PNAS* 2018 (PMID: 30209220), which we used in the present paper.

3. I actually agree that figure 1D,E demonstrate that LRRK2 is not recruited to Galectin-3 positive heavily damaged lysosomes but the comparison between LLOMe and chloroquine is a bit misleading as the time course of lysosomal damage and LRRK2 activation is very different between

these compounds. Authors should use both compounds at different times to show that LRRK2 activation occurs prior to establishment of Gal3+ lysosomes destined for lysophagy.

>> We have re-examined the recruitment of LRRK2 and Galectin-3 during treatment with CQ or LLOMe with the same time course. Although the effect of LLOMe could not be examined beyond 1 hour due to its cytotoxicity, LRRK2 was recruited more efficiently during CQ treatment even at the same time, whereas Galectin-3 was recruited specifically during LLOMe treatment. These data, now shown in Fig 1E, support the notion that LRRK2 recruitment and lysophagy are independent.

4. The CLEM in 1F is also not very convincing, in part because the contrast on the EM portions is very low making it hard to discern organelles and in part because the LRRK2 signal (here with mCherry, which reinforces the concern about endogenous staining) is clearly overexposed, presumably due to that cell having very high levels of transfected protein. I really think better images should be provided, ideally with more comprehensive sampling so we can evaluate the distinction between LRRK2 at lysosomes vs phagosomes.

>> We have now included additional two fields of CLEM images. One of them was shown in Fig. 1F by replacing with the original pictures (which was moved to Fig S1B), and the other was added in Fig S1C. Please note that the cells containing the inset region are those that weakly express mCherry-LRRK2 and are different from the neighboring cells that strongly express mCherry fluorescence.

5. Because of some of the concerns above about staining for LRRK2, and to be able to show robust data, figure 2 recruitment data needs to be validated by at least one orthogonal technique. Lyso-IP or subcellular fractionation would be appropriate.

>> We have adopted an orthogonal biochemical approach using iron-dextran (product name: DexoMAG), where lysosomes internalizing iron-dextran are magnetically isolated and LRRK2 in this fraction was analyzed. As shown in new Fig S2B, LRRK2 was enriched in lysosomal fraction after CQ treatment, and this was further suppressed by siATG5 treatment, validating the results of immunocytochemistry. Successful purification of lysosomes was confirmed by detecting LAMP1 (positive control) and GAPDH (negative control). This method was already described in our previous paper (PMID: 30209220), where additional control proteins were also shown. We avoided performing Lyso-IP because this technique requires overexpression of lysosomal transmembrane protein, which was incompetent in our RAW cells.

6. For each siRNA in figure 2, please include validation that each has the expected knockdown at protein level, particularly for the negative result of FIP200. It would be helpful to show that autophagy initiation complex is rendered non-functional as a result.

>> We have included the data validating efficient knockdown of FIP200 protein levels upon siFIP200 treatment (Fig S3C and Fig 4A). We have also shown that siFIP200 treatment suppressed

starvation-induced autophagy, as shown by LC3 lipidation, confirming the impairment of autophagy initiation complex (Fig S3C). The efficient knockdown of LRRK2 is shown in Fig 2A and Fig 4A, and that of ATG5 is shown in Fig 4A.

7. The major concern with the manuscript as submitted is that the mechanism(s) involved are poorly developed. The interpretation of figures 2 and 3 is that LRRK2 recruitment to stressed lysosomes requires ATG8 conjugation but not the autophagy initiation complex, perhaps instead involving interaction with ATG18L1. Does LRRK2 physically interact with ATG18L1 under conditions of lysosomal stress? The discussion mentions data not presented suggesting not but if this is not true, then how is the ATG8 conjugation machinery involved in LRRK2 recruitment?

>> We understand that this mechanistic aspect is very important. We have tested the interaction between LRRK2 and ATG16L1 (ATG18L1 in this comment likely means ATG16L1) overexpressed in HEK293 cells by immunoprecipitation, and found their interaction. However, this result is not still solid, as pulling down LRRK2 co-precipitated ATG16L1 but the reverse has not been successful so far. Also, if this data is certain, many further questions will arise, e.g., the dependence on ATG12-5-16L1 complex formation and activity, interaction domains, dependence on CQ treatment, etc. So, we have decided not to present this data at this time.

Instead, we now show the results of interaction analysis between LRRK2 and ATG8 ortholog proteins, as this point was also asked by another referee (Reviewer #2, 1st query). Our immunoprecipitation experiments revealed the lack of interaction between LRRK2 and LC3A/B/C or GABARAP/GABARAPL1/GABARAPL2 (Fig S4A and S4B), suggesting that LRRK2 is not recruited to lysosomes via direct binding to ATG8 orthologs. We believe this data is helpful for the understanding of the mechanism.

8. Figure 4A,B are confusing in the context of the rest of the data. Previously, in figure 2E it appeared that siRNA against ATG5 decreased LRRK2 recruitment to lysosomes substantially, from about 60% to 20% of cells. However, in figure 4B the same treatment only has an ~20% decrease in pRAB10 which is statistically significant but clearly much smaller than recruitment. Is LRRK2 still kinase active but away from the lysosome?

>> As mentioned above, we changed the siRNA for ATG5. Since the siRNA we used was a mixture of four different sequences (from Dharmacon), we tested their effect one by one aiming to increase knockdown efficiency. As shown below, siATG5 #1 and #4 had greater effect on both ATG5 and pRab10, whereas siATG5 #2 caused reduction of ATG5 but not pRab10, and siATG5 #3 had little effect. We suspect that #2 has some off-target effects that affect Rab10 dephosphorylation (e.g., phosphatase, Rab-GAP, GDI...), inhibiting the reduction of pRab10. We therefore used siATG5 #1 and repeated Fig 4A experiment. The new Fig 4A now shows clear reduction of pRab10 upon siATG5 treatment, and the quantification was shown in new Fig 4B.

9. A similar concern for 4C,D; if understood correctly this data is from Atg16l1 knockout MEFs but this makes it hard to compare to the data in figure 3E-H which is from Atg16L1 'E320' BMDMs. Both assays should be performed in both genetic models and cell types.

>> Following this suggestion, we assessed CQ-induced phosphorylation of Rab10 using BMDMs derived from WT and E230 mice. The results were as expected, with elevated pRab10 in WT and not in E230. This data is now shown in new Fig 4C and 4D. Original Fig 4C and 4D were moved to Fig 4E and 4F.

10. I am honestly not sure what figure 5A-D tells us. While both knockdown of LRRK2 and ATG5 lowers cathepsin D and B in the media, this is not sufficient to infer that ATG5 affects LRRK2 as the effects could be independent and/or mechanistically unrelated. Evaluation of additive effects of ATG5 and LRRK2 should be performed and mechanism should be developed sufficiently to show that other aspects of Cathepsin biology (synthesis, trafficking, maturation) remain unaffected by both ATG5 and LRRK2 knockdown.

>> It is true that the functional relationship between LRRK2 and ATG5 in cathepsin B/D release is not proven in Fig 5 alone, whereas the regulation of LRRK2 localization and activity by ATG5 under CQ treatment is already shown in Fig 2 and Fig 4. Taken together with our previous results that LRRK2 facilitates cathepsin release via Rab10 and its effectors under CQ treatment (PMID:

30209220), these data suggest that the LRRK2-dependent release is regulated by upstream ATG5. As for the double knockdown of ATG5 and LRRK2, since cathepsin release was already largely suppressed even with a single knockdown, we considered that the additive effect, if any, could not be properly evaluated. We have also shown in Fig 5C that cathepsin B protein levels and the degree of maturation are not affected by knockdown of ATG5 or LRRK2 at steady state.

11. Figure 5E,F is similarly uncertain - there is a non-significant effect of ATG5 knockdown on lysosomal area but the importance of this is obscure. The experiment also does not include siRNA treatments without chloroquine, again raising the concern that these are independent (but not significant) effects.

>> We have repeated this experiment and now found a significant effect of ATG5 knockdown on lysosomal morphology ($p = 0.013$, new Fig 5F). Lysosomes in the absence of CQ were not at all enlarged and remained extremely small even when ATG5 or FIP200 was knocked down, so only immunofluorescence images are included in Fig S5, and their description is given in the text (page 10, lines 260-261). We have also changed the pictures in Fig 5E so that they are consistent with the results in Fig 2C and 2E, following a suggestion from another reviewer (Reviewer #2, 7th query).

Furthermore, we have added LRRK2 knockdown in our quantification (Fig 5E-F), since we realized that our previously reported effect of LRRK2 against lysosomal enlargement (PMID: 30209220) may not be readily understood by the readers, as judged from the comments by other referees (Reviewer #2's 7th query, Reviewer #3's 4th query). We therefore have chosen to include the actual data for LRRK2 knockdown, in addition to the description in the text.

12. RAB29 is invoked as a possible regulator in the discussion but this seems unlikely given that LRRK2 is still activated in RAB29 knockout cells (PMID: 33135724). Given the lack of mechanism in the paper, at least this is a testable idea that should be evaluated experimentally.

>> We have examined the recruitment of endogenous Rab29 onto the enlarged lysosomes upon CQ treatment, which has been shown in our recent paper (PMID: 37365944), and tested whether knockdown of ATG16L1 suppresses Rab29 recruitment. However, unlike LRRK2, Rab29 recruitment was not suppressed by siATG16L1 treatment, suggesting that Rab29 is not a mediator in the recruitment of LRRK2 by the ATG8 conjugation system. These data are now shown in Fig. S4C-D, and the results and discussion are described in page 7-8, lines 187-193 and page 11, lines 285-288, respectively.

Another referee has also asked this point about the role of Rab29 (Reviewer #3, 3rd query), so we have responded in the same way.

Please also note that we still consider that Rab29 is another upstream regulator of LRRK2, given our previous data that transient knockdown of endogenous Rab29 in macrophage cells suppressed LRRK2 activation (PMID:32919031, Fig 6A-B) and LRRK2 recruitment onto the enlarged lysosomes (PMID: 30209220, Fig 3E-F). The difference from the findings in the paper referred to

by this reviewer (PMID: 33135724) is not clear, but the cell types (macrophages vs other types) or experimental conditions (e. g., transient vs stable knockout of Rab29) might have caused the difference.

Reviewer #2 (Comments to the Authors (Required)):

The manuscript by Eguchi and colleagues reports a finding of the ATG8 ubiquitin-like conjugation system that controls the recruitment of Parkinson's disease-related protein LRRK2 and autophagy adaptor LC3 onto single-membrane of stressed lysosome or phagosomes. The recruitment of LRRK2 does not appear to require autophagy initiation regulators (e.g. FIP200). It may involve Atg16L-WD40 domain and perhaps v-ATPase activity. Furthermore, it shows that the ATG8 conjugation regulates LRRK2 activity (as measured with rab10 substrate phosphorylation) and lysosome enzyme release. Despite the interesting findings, the study could benefit from providing molecular mechanism whereby ATG8 conjugation regulates LRRK2 recruitment to stressed lysosomes. The following are my additional comments:

1. The observation seems to suggest that lipidated LC3 (II) may be the link for LRRK2 recruitment to the single membrane of stressed lysosomes. It also remains possible that other ATG8 family member (e.g. GABARAP) could be potential adaptors for the recruitment. It could be also interesting to test if LRRK2 contains conserved LIR, which may mediate the interaction between LRRK2 and ATG8 family members. Additional data from the above study would readily provide important insight and significantly improve the manuscript.

>> Thank you for raising the possibility that ATG8 family proteins may act as adaptors for LRRK2 recruitment. Since LRRK2 has multiple LIR motifs (according to iLIR database), we agreed with this view and examined whether LRRK2 and ATG8 family proteins bind to each other by immunoprecipitation experiments. However, whereas we could detect binding of LC3B to p62, a well-known interactor of LC3, no binding of LRRK2 to ATG8 family proteins was observed. These data are now shown in Fig S4A-B, and their description is given in the result section (page 7, lines 183-187) and discussion section (page 11, lines 282-285).

Regarding other mechanisms of LRRK2 recruitment, we found the possibility of binding of LRRK2 to ATG16L1, as described in response to Reviewer 1's 7th query above. However, as stated above, additional experiments are needed to verify/discuss this binding, so we have decided not to include the data in the current paper.

2. Related to the above point, change of LC3 lipidation status with Atg4 manipulation could be used to test their hypothesis for the purpose of strengthening their conclusion.

>> Thank you for proposing the Atg4 experiment, and we fully agree with this idea. Given the above-mentioned result indicating no binding between LRRK2 and LC3, we sought to confirm whether LRRK2 is recruited even under the suppression of LC3 lipidation by ATG4 inhibition. However, as shown below, treatment with an ATG4B inhibitor NSC185058, which has been shown to suppress LC3 lipidation (PMID: 25483883), at the same concentration and duration as in this paper, resulted in the elevation of LC3 lipidation, while Rab10 phosphorylation decreased. Furthermore, when the four isoforms of ATG4 (ATG4A to 4D) were knocked down individually or in combination, LC3 lipidation was again elevated in some knockdowns, while Rab10 phosphorylation decreased. Since ATG4 is involved not only in the processing of newly synthesized LC3 but also in their delipidation, the result of elevated lipidation is also reasonable, and the lack of correlation between LC3 lipidation and Rab10 phosphorylation may indicate that LC3 is not a mediator of LRRK2 recruitment. However, these data related to ATG4 may further be confusing to the reader, so we decided not to include them in this paper.

3. The author concluded that ATG8 conjugation but not autophagy initiation regulates the LRRK2 and LC3 recruitment. But the claim of "autophagy initiation" is based solely on the assay of FIP200. The study could also include the examination of the component in another autophagy initiation/nucleation regulator complex (Vps34-beclin1-Atg14) to back up the claim.

>> We found that the inhibition of PI3-kinase by treatment with several VPS34 inhibitors markedly suppressed lysosome enlargement caused by CQ, as shown below. Although the recruitment of LRRK2 to LAMP1-positive lysosomes appears to occur even in VPS34-inhibited cells (arrowheads in the pictures below), we thought it would be difficult to quantitatively compare the LRRK2 recruitment in the two groups that differ greatly in lysosomal morphology.

To address this reviewer's concern in a different way, we added a study on ATG13, another component of the autophagy initiation complex containing FIP200. The results were as expected: knockdown of ATG13 suppressed autophagy during starvation induction (Fig S3D) but not LRRK2

recruitment to lysosomes during CQ treatment (Fig S3A-B).

4. The manuscript should include the western blot analysis of various protein levels through siRNA knock-down to demonstrate the reduction of the proteins in their assays. This seems to be a problem for multiple figures.

>> We have now included the immunoblot pictures showing the reduction of protein levels by siRNAs, in Fig 2A (LRRK2), Fig 4A (LRRK2, ATG5, FIP200), Fig S3C (FIP200) and Fig S3D (ATG13).

5. The number of biological repeats shown in figure 2E is inconsistent with the figure legend.

>> We are sorry for this error. The description in the figure legend has been corrected to N = 3.

6. The author should include LRRK2 inhibitor to show the specificity of rab10-phosphorylation antibody (figure 4)

>> It has been well documented in previous studies that Rab10 phosphorylation detected by this antibody is fully suppressed by LRRK2 inhibitor treatment (e.g., PMID:29127256, and our previous paper PMID:32919031). Also, Fig 4A shows that the knockdown of LRRK2 markedly reduced Rab10 phosphorylation. So we believe that the specificity of this antibody is sufficiently assured and widely accepted.

7. Figure 5E shows LRRK2 recruitment to Lamp1+ vesicle in siATG5. This contradicts the previous images of in Figure 2C. Figure 5E also gives the impression that LRRK2- lysosomes are more enlarged than LRRK2+. It would be good to quantify and supplement the idea of LRRK2's importance in lysosome maintenance, especially considering the quantitation in Figure 5F has non-significant change.

>> We have changed the pictures in Fig 5E so that they are consistent with the results in Fig 2C

and 2E. Thank you for pointing out this discrepancy. We have repeated this experiment on lysosomal morphology under the more effective ATG5 knockdown conditions, and now confirmed the statistically significant effect of ATG5 knockdown.

Regarding the effect of LRRK2, we have reported in our previous paper (PMID: 30209220) that the inhibition of LRRK2 using siRNA or small molecule inhibitors markedly enhanced lysosomal enlargement under CQ treatment. These previous data are shown below for reference; we are sorry for not fully explaining this role of LRRK2. The Fig 5E and 5F are intended to show that lysosomal enlargement by CQ is significantly enhanced not only by inhibition of LRRK2 but also by that of ATG5, the latter regulating LRRK2 recruitment and activation from upstream. For the reader's clarity, we have added LRRK2 knockdown in new Fig 5E and 5F.

Fig 5A-B in Eguchi et al. 2018 (PMID: 30209220)

We have also added immunofluorescence images of lysosomes at steady state (i.e., without CQ) in Fig S5, since Reviewer #1 asked whether the effect of ATG5 knockdown on lysosomal enlargement is elicited specifically under CQ treatment. Steady state lysosomes were not at all enlarged by ATG5 knockdown, confirming that the effects of CQ and ATG5 are not parallel.

Reviewer #3 (Comments to the Authors (Required)):

In the manuscript entitled "Non-autophagic ATG8 conjugation to stressed lysosomes recruits LRRK2 to facilitate stress responses", the authors explored the activation mechanism of LRRK2. They found LRRK2 is recruited to single membranes after lysosomal stress. This recruitment is independent of canonical autophagy machinery but relies on p22, Rubicon, and the V-ATPase-ATG16L1 axis. They further demonstrate that these factors are required for the function of LRRK2, which is involved in maintaining lysosomal homeostasis. Through a series of experiments using microscopic and genetic techniques, they provided insights into the role of the V-ATPase-ATG16L1 axis in activating LRRK2. Overall, the manuscript is well-presented, and the figures are organized. However, in its current form, the manuscript is relatively simplified and has several questions that need to be addressed.

1. The title "Non-autophagic ATG8 conjugation to stressed lysosomes recruits LRRK2 to facilitate stress responses" is misleading. In fact, the authors did not directly demonstrate the activation of LRRK2 by non-autophagic ATG8 conjugation. It is possible that the ATG16L1 complex, activated by V-ATPase, can activate LRRK2 directly. Therefore, the authors must examine the LRRK2 phenotype in ATG8-deficient cells, otherwise the title should be changed to "the V-ATPase-ATG16L1 axis".

>> Thank you for pointing out this important issue in this study. We agree that the involvement of the activity of non-autophagic ATG8 conjugation is not directly demonstrated. To generate cells functionally deficient in ATG8, we treated cells with siRNAs or inhibitors for ATG4, a cysteine protease responsible for the processing of newly synthesized ATG8, following the suggestion by another referee (Reviewer #2, 2nd query). However, as stated in the response to Reviewer #2 above, LC3 lipidation was rather enhanced, possibly due to the ATG4's delipidation activity. Although we confirmed in this experiment that the degree of LC3 lipidation and LRRK2 activity are not correlated, we decided not to show this data to avoid confusion of the readers. Following the suggestion of this reviewer, we have changed the title to "The V-ATPase-ATG16L1 axis recruits LRRK2 to facilitate lysosomal stress responses".

2. LAP (p22/Rubicon-dependent) and CASM (V-ATPase/ATG16L1 WD40-dependent and SopF sensitive) are two known non-canonical LC3 activation pathways. In this manuscript, it is surprising that both pathways are required for LRRK2 activation. The authors should explain why activation of LRRK2 requires both LAP and CASM, and what is the connection between these two pathways?

>> To our knowledge, CASM refers to the general phenomenon of "conjugation of ATG8 to endolysosomal single membranes", and LAP induced by phagocytosis is thought to be included in CASM. Although LAP and other CASM have different upstream induction mechanisms, they share the common downstream mechanism of stress to cause lysosomal pH elevation and the consequent activation of the V-ATPase-ATG16L1 axis. Since LRRK2 is regulated downstream of the V-ATPase-ATG16L1 axis, it seems reasonable that LRRK2 is activated both in LAP and other CASM.

3. In the discussion, the authors mentioned the possibility that Rab29 is responsible for the regulation of LRRK2. Did the authors validate the role of Rab29 in their experimental system?

>> Since we have reported that endogenous Rab29 as well as LRRK2 is recruited to the enlarged lysosomes (PMID: 37365944) and that LRRK2 recruitment is suppressed by knocking down of endogenous Rab29 (PMID: 30209220), we examined whether Rab29 recruitment is regulated by the ATG8 conjugation system. We found that the lysosomal recruitment of Rab29 was not suppressed by siATG16L1 treatment, suggesting that Rab29 is not a mediator in the recruitment of LRRK2 by the ATG8 conjugation system. These data are now shown in Fig S4C-D, and the results and discussion are described in page 7-8, lines 187-193 and page 11, lines 285-288, respectively.

Another referee has also asked this point about the role of Rab29 (Reviewer #1, 12th query), so we have responded in the same way.

4. Besides the release of lysosomal cathepsins, are there other assays that indicate lysosomal homeostasis that could corroborate LRRK2 function?

>> As another role of LRRK2 on lysosomal homeostasis, we have reported that LRRK2 acts to suppress lysosomal enlargement driven by CQ (PMID: 30209220). The actual data in this paper were shown in the response to Reviewer #2 above (7th query), since this reviewer also asked about the relationship between LRRK2 and lysosomal enlargement. Fig 5E and 5F are intended to show that not only LRRK2 but also its regulator ATG5 has the same function against lysosomal enlargement, but our explanation was not very clear. To clarify these points, we have now added LRRK2 knockdown in our experiment of lysosomal size (new Fig 5E, 5F).

We have also tried to evaluate other indicators of lysosomal homeostasis, such as changes in lysosomal pH and substrate accumulation in lysosomes, but unfortunately, we have not obtained data suggesting the LRRK2 function related to these changes.

5. The phenotype that ATG5/ATG16L1 regulates the kinase activity of LRRK2 is relatively weak in Fig. 4. The authors could emphasize this conclusion by detection of the level of phosphorylated Rab8, and by overexpression of SopF instead of ATG5 knockdown.

>> In Fig 4A-B, we have re-examined the knockdown of ATG5 using different siRNA, as another referee also asked for an explanation for the weak phenotype of pRab10 reduction (Reviewer #1, 8th query). The detail of how we changed the siRNA is described in our response to Reviewer #1; we now confirmed that ATG5 knockdown greatly reduced Rab10 phosphorylation, as shown in the new Fig 4A-B.

Regarding the effect of ATG16L1, the original Fig 4C-D (now Fig 4E-F) show that the knockout almost completely suppressed pRab10 elevation by CQ, so we think that the phenotype was already robust.

Regarding Rab8 phosphorylation, we avoided testing this because a commercially available antibody for phosphorylated Rab8 (at Thr72) is known to cross-react with phosphorylated Rab3A, Rab10, Rab35 and Rab43, as described in its datasheet (abcam, MJF-R20).

Regarding the experiment of SopF overexpression, we have instead performed the experiment using bone marrow-derived macrophages (BMDMs) from WT and E230 mice, the latter lacking V-ATPase-interacting region of ATG16L1. As shown in new Fig 4C-D, we confirmed that E230 BMDMs failed to show an increase in phospho-Rab10 upon CQ exposure.

We believe that these data altogether support the conclusion that the non-autophagic function of the ATG8 conjugation system, especially the V-ATPase-ATG16L1 axis, regulates LRRK2 kinase activity.

November 28, 2023

RE: JCB Manuscript #202302067R

Dr. Tomoki Kuwahara
The University of Tokyo
Dept. Neuropathology, Grad. Sch. Med.
7-3-1, Hongo, Bunkyo-ku
Tokyo 1130033
Japan

Dear Dr. Kuwahara:

Thank you for submitting your revised manuscript entitled "The V-ATPase-ATG16L1 axis recruits LRRK2 to facilitate lysosomal stress responses". We appreciate your patience as we gathered reviewer input which required longer than usual due to staff availability. We would be happy to publish your paper in JCB pending final revisions necessary to meet our formatting guidelines (see details below).

A. MANUSCRIPT ORGANIZATION AND FORMATTING:

Full guidelines are available on our Instructions for Authors page, <http://jcb.rupress.org/submission-guidelines#revised>. Submission of a paper that does not conform to JCB guidelines will delay the acceptance of your manuscript.

- 1) Text limits: Character count for Articles is < 40,000, not including spaces. Count includes abstract, introduction, results, discussion, and acknowledgments. Count does not include title page, figure legends, materials and methods, references, tables, or supplemental legends.
- 2) Figures limits: Articles may have up to 10 main figures and 5 supplemental figures/tables.
- 3) Figure formatting: Scale bars must be present on all microscopy images, including inset magnifications. Molecular weight or nucleic acid size markers must be included on all gel electrophoresis. Please avoid pairing red and green for images and graphs to ensure legibility for color-blind readers. If red and green are paired for images, please ensure that the particular red and green hues used in micrographs are distinctive with any of the colorblind types. If not, please modify colors accordingly or provide separate images of the individual channels.
- 4) Statistical analysis: Error bars on graphic representations of numerical data must be clearly described in the figure legend. The number of independent data points (n) represented in a graph must be indicated in the legend. Statistical methods should be explained in full in the materials and methods. For figures presenting pooled data the statistical measure should be defined in the figure legends. Please also be sure to indicate the statistical tests used in each of your experiments (either in the figure legend itself or in a separate methods section) as well as the parameters of the test (for example, if you ran a t-test, please indicate if it was one- or two-sided, etc.). Also, if you used parametric tests, please indicate if the data distribution was tested for normality (and if so, how). If not, you must state something to the effect that "Data distribution was assumed to be normal but this was not formally tested."
- 5) Abstract and title: The abstract should be no longer than 160 words and should communicate the significance of the paper for a general audience. The title should be less than 100 characters including spaces. Make the title concise but accessible to a general readership.
** For clarity we recommend changing the title to:
"The V-ATPase-ATG16L1 axis recruits LRRK2 to facilitate the lysosomal stress response."
- 6) Materials and methods: Should be comprehensive and not simply reference a previous publication for details on how an experiment was performed. Please provide full descriptions in the text for readers who may not have access to referenced manuscripts. We also provide a report from SciScore and an associate score, which we encourage you to use as a means of evaluating and improving the methods section.
- 7) Please be sure to provide the sequences for all of your primers/oligos and RNAi constructs in the materials and methods. You must also indicate in the methods the source, species, and catalog numbers (where appropriate) for all of your antibodies. Please also indicate the acquisition and quantification methods for immunoblotting/western blots.

8) Microscope image acquisition: The following information must be provided about the acquisition and processing of images:

- Make and model of microscope
- Type, magnification, and numerical aperture of the objective lenses
- Temperature
- Imaging medium
- Fluorochromes
- Camera make and model
- Acquisition software
- Any software used for image processing subsequent to data acquisition. Please include details and types of operations involved (e.g., type of deconvolution, 3D reconstitutions, surface or volume rendering, gamma adjustments, etc.).

10) Supplemental materials: There are strict limits on the allowable amount of supplemental data. Articles may have up to 5 supplemental figures. Please also note that tables, like figures, should be provided as individual, editable files. A summary of all supplemental material should appear at the end of the Materials and methods section.

13) ORCID IDs: ORCID IDs are unique identifiers allowing researchers to create a record of their various scholarly contributions in a single place. At resubmission of your final files, please add an ORCID ID for all authors.

Please note that JCB now requires authors to submit Source Data used to generate figures containing gels and Western blots with all revised manuscripts. This Source Data consists of fully uncropped and unprocessed images for each gel/blot displayed in the main and supplemental figures. Since your paper includes cropped gel and/or blot images, please be sure to provide one Source Data file for each figure that contains gels and/or blots along with your revised manuscript files. File names for Source Data figures should be alphanumeric without any spaces or special characters (i.e., SourceDataF#, where F# refers to the associated main figure number or SourceDataFS# for those associated with Supplementary figures). The lanes of the gels/blots should be labeled as they are in the associated figure, the place where cropping was applied should be marked (with a box), and molecular weight/size standards should be labeled wherever possible. Source Data files will be directly linked to specific figures in the published article.

Journal of Cell Biology now requires a data availability statement for all research article submissions. These statements will be published in the article directly above the Acknowledgments. The statement should address all data underlying the research presented in the manuscript. Please visit the JCB instructions for authors for guidelines and examples of statements at (<https://rupress.org/jcb/pages/editorial-policies#data-availability-statement>).

B. FINAL FILES:

-- Cover images: If you have any striking images related to this story, we would be happy to consider them for inclusion on the journal cover. Submitted images may also be chosen for highlighting on the journal table of contents or JCB homepage carousel.

Images should be uploaded as TIFF or EPS files and must be at least 300 dpi resolution.

****It is JCB policy that if requested, original data images must be made available to the editors. Failure to provide original images upon request will result in unavoidable delays in publication. Please ensure that you have access to all original data images prior to final submission.****

****The license to publish form must be signed before your manuscript can be sent to production. A link to the electronic license to publish form will be sent to the corresponding author only. Please take a moment to check your funder requirements before choosing the appropriate license.****

Thank you for this interesting contribution, we look forward to publishing your paper in Journal of Cell Biology.

Sincerely,

Hong Zhang
Monitoring Editor
Journal of Cell Biology

Tim Fessenden
Scientific Editor
Journal of Cell Biology

Reviewer #1 (Comments to the Authors (Required)):

I am satisfied with the authors' responses.

Reviewer #2 (Comments to the Authors (Required)):

The authors have adequately addressed my concerns and I have no further questions.

Reviewer #3 (Comments to the Authors (Required)):

The picture quality and rigor of this article have significantly improved in this version, and the author has addressed most of my inquiries. Although some questions remain, such as the direct binding of ATG16L1 to activate LRRK2, the paper presents a novel function of the noncanonical autophagy pathway. I recommend its publication in JCB magazine.